# Epigenetic age acceleration, telomere length, and neurocognitive function in long-term survivors of childhood cancer

AnnaLynn M. Williams [1,2] ✉, Nicholas S. Phillips [3], Qian Dong[1], Matthew J. Ehrhardt [1,4], Nikesha Gilmore[2], Kah Poh Loh [5], Xiaoxi Meng[1], Kirsten K. Ness [1], Melissa M. Hudson[1], Leslie L. Robison[1], Zhaoming Wang [1,6] & Kevin R. Krull [3,6] ✉

Survivors of childhood cancer are prone to neurocognitive impairment and premature aging, raising concerns about early onset dementia. In this cross-sectional study, 1413 survivors of childhood cancer complete a neuropsychological battery. Mean leukocyte telomere length residual (mLTL) and epigenetic age acceleration (EAA) from five different epigenetic clocks, are derived from linear regression of mLTL or epigenetic age on chronological age. Among survivors treated with CNS-directed therapy, higher EAA, measured by PCGrimAge, or DunedinPACE is associated with worse performance on multiple measures of attention, processing speed, and executive functions (p's < 0.05). Among non-CNS-treated survivors, results are similar for PCGrimAge, however, DunedinPACE is specifically associated with attention variability (p < 0.05). mLTL is not associated with neurocognition. EAA is associated with worse neurocognitive function and may identify survivors at risk for accelerated cognitive aging or serve as an efficacy biomarker for neurocognitive interventions.

Currently, more than 500,000 survivors of childhood cancer live in the United States, over 40% of whom will experience persistent neuro-cognitive impairment years after treatment[1,2]. Common neurocognitive domains affected include attention, memory, executive function, and processing speed, which are associated with lower quality of life, lower educational attainment, and deficits in independent living, social attainment, and employment[3]. While it is clear that patients treated with central nervous system (CNS) directed therapies are at risk for brain abnormalities, emerging evidence indicates other treatments, like non-CNS radiation and DNA-damaging chemotherapies, may pose a significant risk to the CNS[2,4]. For example, despite receiving no CNS-directed therapies, long-term survivors of Hodgkin lymphoma (HL) are

more likely to be impaired on measures of attention and memory[5,6], and are at higher risk of stroke[7], compared to siblings, both of which are associated with dementia.

Childhood cancer survivors are exposed to various non-CNS-directed treatments that can cause molecular damage, changing DNA structure, cellular function, signaling, and tissue integrity[8,9]. These exposures happen during key developmental stages and may alter growth and maturation, and subsequently set survivors on a unique physiologic, biologic, and cognitive aging trajectory. Accumulation of molecular damages over time may result in increased inflammation, oxidative stress, immunosenescence, telomere attrition (e.g., mean leukocyte telomere length), and epigenetic modifications (e.g.,

[1]Department of Epidemiology and Cancer Control, St. Jude Children's Research Hospital, Memphis, TN, USA. [2]Department of Surgery, University of Rochester School of Medicine and Dentistry, Rochester, NY, USA. [3]Department of Psychology and Biobehavioral Sciences, St. Jude Children's Research Hospital, Memphis, TN, USA. [4]Department of Oncology, St. Jude Children's Research Hospital, Memphis, TN, USA. [5]Department of Medicine, University of Rochester School of Medicine and Dentistry, Rochester, NY, USA. [6]These authors contributed equally: Zhaoming Wang, Kevin R. Krull. ✉ e-mail: AnnaLynn_Williams@urmc.rochester.edu; Kevin.Krull@St.Jude.org

epigenetic age acceleration (EAA)) earlier than expected[10]. Childhood cancer survivors have shorter telomere length and increased EAA compared to non-cancer controls, manifesting premature cellular aging[11,12]. These well-established aging-related biomarkers have previously been associated with neurocognitive impairment and decline in older non-cancer populations, particularly in cognitive domains related to aging and dementia, such as memory, attention, and executive function, but these associations have yet to be examined among younger survivors of childhood cancer[13–24].

Several studies have demonstrated that chemotherapy and radiation can impart significant DNA methylation changes in the context of breast cancer and that these changes are associated with functional and cognitive impairments[25–27]. Importantly, epigenetic changes were sustained two years post-therapy, suggesting that epigenetic alterations from cancer treatment persist into long-term survivorship[27,28]. Given the vast number of CpG sites affected, epigenetic clocks may serve as a way to synthesize these data into one composite measure of biological age. At present, several epigenetic clocks have been created, with the first generation of clocks trained to predict chronological age (e.g., Horvath[29], Hannum[30]), the second generation of clocks trained to predict morbidity and mortality (e.g., Levine's PCPhenoAge[31,32], PCGrimAge[32]), and the third generation focused on predicting the pace of aging (e.g., DunedinPACE[33]). Sehl and colleagues have reported significant increases in various epigenetic clocks from pre- to post-chemotherapy and radiation treatment for breast cancer[34]. Similarly, findings from the Thinking and Living with Cancer study suggest increases in epigenetic age are sustained 2–3 years post-breast cancer treatment and are associated with functional and cognitive impairments[26]. These findings highlight that epigenetic modifications from cancer treatment may extend well into survivorship, consistent with our cross-sectional work in long-term survivors of childhood cancer that suggests they experience significant EAA 10 or more years from treatment[11,12,35]. However, the association between epigenetic modifications and long-term neurocognitive impairment has yet to be examined among survivors of childhood cancer who may be experiencing premature cognitive aging[3,36,37].

Here, we show that EAA, but not telomere length (mLTL), is associated with worse neurocognitive function. Analyses are stratified by receipt of CNS-directed therapies as they may increase vulnerability to aging-related changes in cognitive function due to diminished cognitive reserve[2,38]. These data may help to identify survivors at risk for accelerated cognitive aging or serve as an efficacy biomarker for neurocognitive interventions. Additionally, given that epigenetic changes are modifiable, understanding the relationship between biologic and cognitive aging may inform targeted interventions to mitigate neurocognitive limitations in long-term survivors of childhood cancer.

## Results
A total of 1413 adult survivors of childhood cancer and 282 non-cancer controls from the St. Jude Lifetime Cohort (SJLIFE) had available biospecimens and neurocognitive data from the same time point in long-term survivorship and were included in this analysis. Survivors were, on average, 26 years from diagnosis and did not differ significantly in age at evaluation from the non-cancer controls (Table 1). Analyses were stratified by receipt of CNS-directed therapy during treatment for childhood cancer as previous research has demonstrated this is associated with increased vulnerability to aging-related changes due to diminished cognitive reserve[2,39]. The CNS-treated group was comprised largely of acute lymphoblastic leukemia survivors (77%), while the non-CNS-treated group had high proportions of HL (32%) and Wilms tumor (11%) survivors. We first compared survivors in our analytic sample to the remaining survivors in SJLIFE to examine any potential for selection bias. Compared to non-CNS-treated

survivors without biomarker data, those with biomarker data were more likely to have received non-CNS-directed radiation (Supplementary Table 1). Similarly, CNS-treated survivors with biomarker data were less likely to have had CNS-directed radiation and high-dose cytarabine compared to those without biomarker data.

## Measures of biologic aging in survivors and controls
We examined the association between neurocognitive function and biologic aging among long-term survivors of childhood cancer using mLTL and several epigenetic clocks as it is currently unclear which epigenetic clock may best predict neurocognitive dysfunction in this population. PCPhenoAge was initially chosen as our primary outcome because its original derivation, DNAmPhenoAge, outperforms other epigenetic clocks in terms of prediction of all-cause mortality, and age-related health outcomes in the general population and has been recently used to demonstrate accelerated aging in childhood cancer survivors[11,29,32,35,40]. We examined four other epigenetic clocks (PCGrimAge[32], DunedinPACE[33], Horvath[29] Hannum[30]) that have also recently been demonstrated to characterize accelerated aging in childhood cancer survivors[40]. Notably, DunedinPACE was developed among a sample of young adults and may align better to our sample of young adult survivors of childhood cancer. We used the principal components (PC) version of DNAmPhenoage (and GrimAge) to reduce noise due to technical variability of individual DNA methylation sites[32]. For each clock (except DunedinPACE), we derived the EAA by calculating the residual for each individual based on the linear regression model of epigenetic age against age at blood draw for DNA sampling.

Similar to previously published data from the SJLIFE cohort[11], our subsample of survivors experienced significantly higher EAA on all epigenetic clocks and shortened mLTL when compared to non-cancer controls, even after adjustment for relevant covariables such as sex, BMI, physical activity, and smoking(Fig. 1 and Supplementary Table 2). The EAA difference between survivors vs. non-cancer controls appeared larger in non-CNS-treated survivors than CNS-treated survivors. Further, non-CNS-treated survivors experienced significantly greater EAA than CNS-treated survivors when using PCPhenoAge, Horvath, or Hannum epigenetic clocks (Supplementary Table 2).

## Associations between biologic aging and cognition among non-CNS-treated survivors
All survivors completed a comprehensive neurocognitive assessment at the time of biospecimen collection (Supplementary Table 3). To increase interpretability, tertiles of each biomarker were generated based on all samples for use in linear regression models that estimated mean differences in age-adjusted neurocognitive z-scores across tertiles of EAA/mLTL, adjusting for sex, age at diagnosis, body mass index (BMI), smoking, physical activity, and treatments associated with worse neurocognitive impairment. Because this is the first study to examine epigenetic aging and neurocognitive function among long-term survivors of childhood cancer, we did not have a priori hypotheses about which domains and which neuropsychological assessments would be most affected. Therefore, we chose to be conservative and adjust p-values for multiple comparisons using the false discovery rate[41].

Among non-CNS-treated survivors, those in the third tertile of PCPhenoAge EAA performed 0.19 standard deviations worse on a test of visual-motor processing speed compared to those in the first tertile (Table 2, $\beta = -0.19$ 95% CI: $-0.36$, $-0.02$; $p = 0.049$). The third tertile was also associated with worse non-verbal reasoning ($\beta = -0.22$ 95% CI: $-0.36$, $-0.07$; $p = 0.013$) compared to the first tertile. After adjusting for education, these associations attenuated slightly (Supplementary Table 4).

When PCGrimAge was used, associations between EAA and neurocognitive dysfunction were noted across all domains (Table 3). For example, the third tertile of PCGrimAge EAA was associated with a

**Table 1 | Participant characteristics**

| | Non-CNS-treated survivors (*n* = 663) | CNS-treated survivors (*n* = 750) | *p*-value* | Community controls (*n* = 282) | *p*-value* |
|---|---|---|---|---|---|
| | Mean (SD) | Mean (SD) | | Mean (SD) | |
| Age at evaluation (years) | 36.42 (9.79) | 34.10 (8.76) | <0.001 | 35.85 (10.25) | 0.286 |
| Age at diagnosis (years) | 9.93 (6.46) | 7.30 (4.82) | <0.001 | | |
| Time since diagnosis (years) | 26.49 (9.41) | 26.80 (8.60) | 0.523 | | |
| BMI | 28.81 (7.56) | 30.84 (7.75) | <0.001 | 29.01 (7.51) | 0.082 |
| Physical activity (Met h/wk) | 7.34 (11.05) | 6.95 (10.32) | 0.494 | 7.27 (8.56) | 0.814 |
| | N(%) | N(%) | | N(%) | |
| Male sex | 352 (53.1) | 395 (52.7) | 0.873 | 137 (48.6) | 0.189 |
| Female sex | 311 (46.9) | 355 (47.3) | | 145 (51.4) | |
| Education[a] | | | 0.099 | | <0.001 |
| <College | 349 (56.9) | 432 (62.5) | | 118 (45.0) | |
| ≥College Graduate | 270 (43.1) | 259 (37.5) | | 144 (55.0) | |
| Current Smoker | 140 (21.1) | 150 (20.0) | 0.002 | 36 (12.8) | 0.011 |
| Comorbidities[b] | | | | | |
| None/Low | 213 (32.1) | 228 (30.4) | 0.851 | 143 (50.7) | <0.0001 |
| Moderate | 166 (25.0) | 186 (24.8) | | 80 (28.4) | |
| High | 211 (31.8) | 254 (33.9) | | 55 (19.5) | |
| Severe | 73 (11.0) | 82 (10.9) | | 4 (1.4) | |
| Diagnosis | | | | | |
| Acute lymphoblastic leukemia | 1 (0.2) | 577 (76.9) | | | |
| Hodgkin lymphoma | 211 (31.8) | 5 (0.7) | | | |
| Neuroblastoma | 60 (9.1) | 0 (0.0) | | | |
| Non-Hodgkin lymphoma | 28 (4.2) | 105 (14.0) | | | |
| Osteosarcoma | 59 (8.9) | 2 (0.3) | | | |
| Other | 231 (34.8) | 60 (8.0) | | | |
| Wilms tumor | 73 (11.0) | 1 (0.1) | | | |
| Radiation | | | | | |
| No radiation treatment | 242 (36.5) | 334 (44.5) | 0.002 | | |
| Brain radiation (yes) | - | 379 (26.8) | - | | |
| >0–30 Gy | - | 163 (35.0) | | | |
| >30 Gy | - | 116 (15.5) | | | |
| Chest (yes) | 306 (46.2) | 76 (10.1) | <0.001 | | |
| Abdomen/Pelvic (yes) | 209 (31.5) | 70 (9.3) | <0.001 | | |
| Chemotherapy (yes) | | | | | |
| High-dose IV cytarabine | 5 (0.8) | 70 (9.3) | <0.001 | | |
| High-dose IV methotrexate | 60 (9.1) | 396 (52.8) | <0.001 | | |
| Intrathecal methotrexate | - | 697 (92.9) | - | | |
| Vincristine | 378 (57.0) | 718 (95.7) | <0.001 | | |
| Anthracyclines | 409 (61.7) | 549 (73.2) | <0.001 | | |
| Cyclophosphamide | 409 (61.7) | 515 (68.7) | 0.006 | | |
| Platinum agents | 105 (15.8) | 20 (2.7) | <0.001 | | |

*CNS* central nervous system, *SD* standard deviation, *BMI* body mass index, *Met* metabolic equivalent, *GED* graduate educational development, *Gy* gray, *IV* intravenous.

[a]103 survivors and 20 controls missing education.

[b]Chronic health conditions were clinically evaluated and systematically graded according to modified NCI Common Terminology Criteria for Adverse Events[78]. Thirty-eight different composite groups of chronic health conditions were used to define comorbidity severity burden score according to previously published methods[2,79]. Categories were defined as "none/low" being grade 1 conditions only; "medium" being ≥1 grade 2 and/or 1 grade 3 condition(s); "high" being ≥2 grade 3, or 1 grade 4 and 1 grade 3 conditions; and "very high" being ≥2 grade 4 or ≥2 grade 3 and 1 grade 4 condition(s),

*Two sample *t*-tests or chi-square tests.

third of a standard deviation (SD) worse performance on attention variability ($\beta = -0.32$ 95% CI: −0.54, −0.09; $p = 0.017$), visual-motor processing speed ($\beta = -0.39$ 95% CI: −0.57, −0.21; $p < 0.001$), memory span ($\beta = -0.33$ 95% CI: −0.53, −0.14, $p = 0.003$), and working memory ($\beta = -0.33$ 95% CI: −0.51, −0.15, $p = 0.002$) compared to the first tertile. After further adjustment for education, effect sizes attenuated slightly, but statistically significant associations remained for visual-motor

processing speed, working memory, and verbal fluency (Supplementary Table 5).

Higher tertiles of DunedinPACE were associated with a third of a standard deviation worse performance on a test of attention variability ($\beta = -0.30$ 95% CI: −0.52, −0.07; $p = 0.038$, Table 4) compared to those in the first tertile. Higher DunedinPACE was also associated with worse performance on measures of global cognition and academics. After

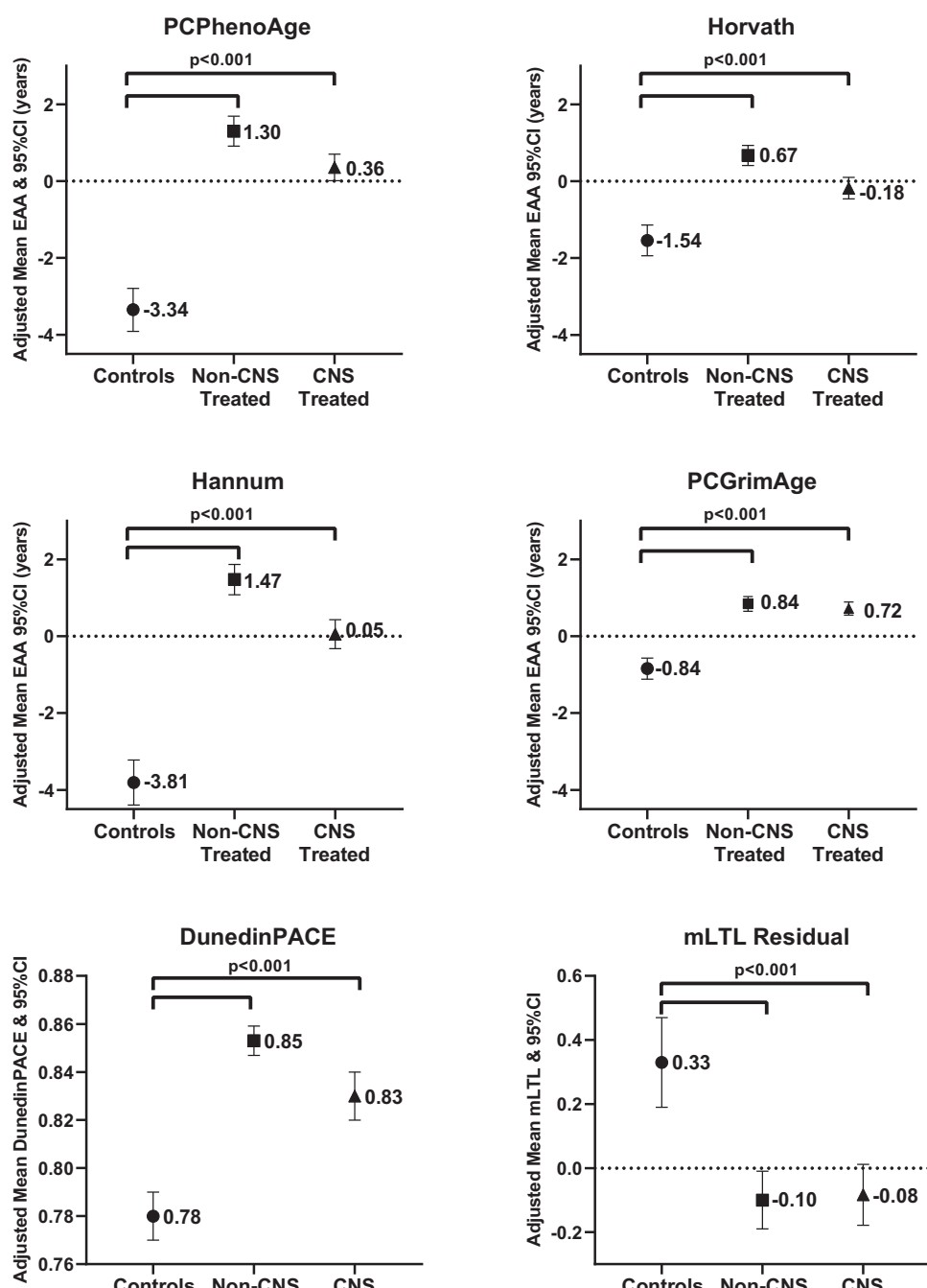

**Fig. 1 | Epigenetic age acceleration (EAA) in survivors of childhood cancer treated with (*n* = 750) and without (*n* = 633) central nervous system directed therapy compared with non-cancer controls (*n* = 282).** Marginal adjusted means and 95%CI (error bars) from linear regression models (*t*-test, two-sided, no adjustment for multiple comparisons) adjusted for sex, body mass index, smoking, and physical activity. Regardless of exposure to CNS-directed therapy, survivors experienced significantly higher EAA, faster DundedinPACE, and shorter mean leukocyte telomere length compared with non-cancer controls (all *p* < 0.001), full results can be found in Supplementary Table 2. Source data are provided as a Source data file.

adjustment for education, associations attenuated slightly (Supplementary Table 6). Lastly, higher tertiles of the Hannum EAA were associated with worse performance on tests of word reading, however, after adjustment for education associations were no longer noted (Supplementary Table 7). No statistically significant associations between mLTL residual or the Horvath epigenetic clock and neurocognitive outcomes were noted after adjustment for multiple comparisons (Supplementary Tables 8 and 9).

We conducted sensitivity analyses among survivors of HL, because they experience a disproportionately high amount of EAA compared to other diagnoses (Supplementary Table 10)[11]. Among HL, higher PCPhenoAge EAA was associated with worse memory span, although not statistically significant after adjustment for multiple comparisons ($\beta = -0.58$ 95% CI: $-1.05, -0.11$; $p = 0.063$ Supplementary Table 10) as was DunedinPACE ($\beta = -0.47$ 95% CI: $-0.87, -0.08$; $p = 0.075$, Supplementary Table 11) and PCGrimAge ($\beta = -0.66$ 95% CI: $-1.01, -0.30$, $p = 0.001$). Higher PCGrimAge EAA was also associated with worse performance in tests of visual-motor processing speed ($\beta = -0.58$ 95% CI: $-0.90, -0.26$; $p = 0.001$), new verbal encoding ($\beta = -0.51, -0.88, -0.14$; $p = 0.012$), working memory ($\beta = -0.54, -0.86$,

**Table 2 | Mean difference (95% CI) in neurocognitive *z*-score associated with the second or third tertile of PCPhenoAge EAA compared to the first tertile stratified by exposure to CNS-directed therapy**

| | Non-CNS-treated[a] *n* = 663 | | | | CNS-treated[b] *n* = 750 | | | |
|---|---|---|---|---|---|---|---|---|
| | Tertile 2 vs. 1 | | Tertile 3 vs. 1 | | Tertile 2 vs. 1 | | Tertile 3 vs. 1 | |
| | *β* (95% CI) | *p*[*] | *β* (95% CI) | *p*[*] | *β* (95% CI) | *p*[*] | *β* (95% CI) | *p*[*] |
| Global cognition | | | | | | | | |
| Verbal reasoning | 0.01 (−0.19, 0.21) | 0.973 | −0.13 (−0.33, 0.07) | 0.199 | −0.15 (−0.33, 0.04) | 0.210 | −0.29 (−0.49, −0.09) | 0.013 |
| Non-verbal reasoning | −0.03 (−0.17, 0.12) | 0.973 | −0.22 (−0.36, −0.07) | 0.013 | −0.07 (−0.23, 0.09) | 0.395 | −0.18 (−0.35, −0.01) | 0.051 |
| Academics | | | | | | | | |
| Word reading | −0.002 (−0.11, 0.11) | 0.973 | −0.09 (−0.20, 0.02) | 0.129 | −0.09 (−0.21, 0.02) | 0.210 | −0.17 (−0.29, −0.05) | 0.013 |
| Mathematics | −0.10 (−0.27, 0.07) | 0.973 | −0.16 (−0.33, 0.01) | 0.129 | −0.13 (−0.30, 0.05) | 0.210 | −0.17 (−0.36, 0.01) | 0.068 |
| Attention | | | | | | | | |
| Sustained | 0.13 (−0.08, 0.34) | 0.478 | −0.15 (−0.36, 0.06) | 0.316 | −0.05 (−0.27, 0.16) | 0.799 | −0.14 (−0.37, 0.09) | 0.325 |
| Variability | 0.001 (−0.21, 0.21) | 0.990 | −0.18 (−0.39, 0.03) | 0.316 | −0.10 (−0.29, 0.10) | 0.799 | −0.16 (−0.37, 0.05) | 0.325 |
| Commissions | 0.09 (−0.11, 0.28) | 0.520 | −0.07 (−0.26, 0.13) | 0.504 | 0.06 (−0.13, 0.25) | 0.799 | −0.08 (−0.28, 0.11) | 0.408 |
| Focused attention | −0.12 (−0.31, 0.08) | 0.478 | −0.10 (−0.29, 0.09) | 0.406 | −0.03 (−0.23, 0.18) | 0.799 | −0.14 (−0.35, 0.08) | 0.325 |
| Processing Speed | | | | | | | | |
| Visual-motor | 0.01 (−0.16, 0.18) | 0.888 | −0.19 (−0.36, −0.02) | 0.049 | −0.07 (−0.24, 0.11) | 0.808 | −0.10 (−0.28, 0.08) | 0.590 |
| Motor | 0.07 (−0.15, 0.28) | 0.888 | −0.01 (−0.22, 0.20) | 0.921 | 0.03 (−0.18, 0.23) | 0.808 | −0.02 (−0.23, 0.20) | 0.867 |
| Memory | | | | | | | | |
| Span | 0.17 (−0.02, 0.36) | 0.289 | −0.14 (−0.32, 0.05) | 0.560 | −0.08 (−0.25, 0.09) | 0.350 | −0.10 (−0.28, 0.08) | 0.284 |
| New verbal encoding | 0.05 (−0.16, 0.25) | 0.850 | −0.06 (−0.26, 0.14) | 0.771 | −0.14 (−0.34, 0.06) | 0.248 | −0.20 (−0.42, 0.01) | 0.234 |
| Short-term verbal recall | −0.10 (−0.30, 0.10) | 0.649 | −0.10 (−0.30, 0.09) | 0.579 | −0.13 (−0.32, 0.06) | 0.248 | −0.14 (−0.34, 0.07) | 0.257 |
| Long-term verbal recall | −0.01 (−0.21, 0.19) | 0.920 | −0.03 (−0.23, 0.17) | 0.784 | −0.16 (−0.36, 0.05) | 0.248 | −0.18 (−0.40, 0.04) | 0.234 |
| Executive function | | | | | | | | |
| Perseveration | 0.13 (−0.10, 0.35) | 0.801 | −0.14 (−0.36, 0.09) | 0.226 | 0.01 (−0.23, 0.24) | 0.964 | −0.01 (−0.26, 0.25) | 0.968 |
| Working memory | 0.08 (−0.10, 0.25) | 0.801 | −0.14 (−0.31, 0.04) | 0.226 | 0.03 (−0.13, 0.19) | 0.915 | −0.07 (−0.24, 0.10) | 0.540 |
| Cognitive switching/flexibility | 0.004 (−0.24, 0.25) | 0.973 | −0.19 (−0.44, 0.05) | 0.226 | −0.19 (−0.45, 0.07) | 0.584 | −0.27 (−0.55, 0.00) | 0.207 |
| Verbal fluency | 0.01 (−0.20, 0.23) | 0.973 | −0.13 (−0.34, 0.07) | 0.226 | −0.06 (−0.24, 0.12) | 0.915 | −0.15 (−0.35, 0.04) | 0.232 |

*EAA* epigenetic age acceleration calculated used PCPhenoAge epigenetic clock.
[a]Models are adjusted for sex, diagnosis age, BMI, physical activity, smoking, and high-dose IV methotrexate.
[b]Models are adjusted for sex, diagnosis age, BMI, physical activity, smoking, cranial radiation dose, intrathecal methotrexate, high-dose IV methotrexate, high-dose cytarabine, and neurosurgery.
[*]Adjusted for multiple comparisons using the false-discovery rate.

−0.22; *p* = 0.003) and verbal fluency (β = −0.53 95% CI: −0.93, −0.13; *p* = 0.018).

**Associations between biologic aging and cognition among CNS-treated survivors**
Among CNS-treated survivors, those in the third tertile of PCPhenoAge EAA performed worse on tests of verbal reasoning (β = −0.29 95% CI: −0.49, −0.09; *p* = 0.013) and word reading (β = −0.17 95% CI: −0.29, −0.05; *p* = 0.013 Table 2). Associations were lost after adjustment for education (Supplementary Table 4).

When PCGrimAge was used as the measure of EAA, statistically significant associations were noted for each cognitive domain across both tertile 2 and tertile 3, with effect sizes increasing as tertile of EAA increased (Table 3). For example, those in the second tertile of PCGrimAge EAA performed a quarter of a SD worse, on average, on a test of long-term recall compared to tertile 1, while those in tertile 3 performed 0.44 SD worse, on average, compared to tertile 1 (tertile 2 β = −0.25 95% CI: −0.46, −0.03; *p* = 0.030, tertile 3 β = −0.44 95% CI: −0.68, −0.20; *p* = 0.001). Similar associations across tertiles were noted for tests of sustained attention, attention variability, visual-motor and motor processing speed, memory span, and working memory (*p* < 0.05). After adjustment for education, associations attenuated but remained statistically significant for attention variability, visual-motor and motor processing speed, short-term recall, working memory, and verbal fluency (Supplementary Table 5).

The third tertile of DunedinPACE was associated with significantly worse performance on tests across all cognitive domains (Table 4).

Those in the third tertile performed over a half of a SD worse on a test of cognitive flexibility compared to those in the first tertile (β = −0.58 95% CI: −0.88, −0.29; *p* < 0.001). Interestingly, both those in the second and third tertile performed 0.3–0.4 SD worse than those in the first on tests of memory, including new verbal encoding, short-term verbal recall, and long-term verbal recall with a similar effect size across tertiles, suggesting a threshold effect may be reached at the second tertile. After adjustment for education, the association with cognitive flexibility remained (β = −0.37 95% CI −0.65, −0.09; *p* = 0.036), as did associations with long-term verbal recall (β = −0.31 95% CI −0.55, −0.08; *p* = 0.037) and visual-motor processing speed (β = −0.23 95% CI −0.42, −0.04; *p* = 0.031 Supplementary Table 6). There were no associations between Horvath or Hannum Clocks, or mLTL with neurocognitive function with the exception of worse verbal reasoning and mLTL (Supplementary Table 7 through 9).

Sensitivity analyses were conducted among survivors of ALL only, as they represented the majority of the sample of CNS-treated survivors. Associations between PCPhenoAge EAA and verbal reasoning or word reading were similar and remained statistically significant (Supplementary Table 12). The third tertile of DunedinPACE and PCGrimAge were associated with significantly worse performance across all neurocognitive assessments, with effect sizes similar to the entire CNS-treated group (Supplementary Table 13). Previous research suggests that age at diagnosis and treatment can impact survivor's vulnerability to short- and long-term cognitive impairment[42]. Therefore, we examined if there was a statistical interaction between age at diagnosis and EAA. There were no statistically significant interactions

**Table 3 | Mean difference (95% CI) in neurocognitive *z*-score associated with the second or third tertile of PCGrimAge EAA compared to the first tertile stratified by exposure to CNS-directed therapy.5**

| | Non-CNS-treated[a] *n* = 663 | | | | CNS-treated[b] *n* = 750 | | | |
|---|---|---|---|---|---|---|---|---|
| | Tertile 2 vs. 1 | | Tertile 3 vs. 1 | | Tertile 2 vs. 1 | | Tertile 3 vs. 1 | |
| | *β* (95% CI) | *p*[*] | *β* (95% CI) | *p*[*] | *β* (95% CI) | *p*[*] | *β* (95% CI) | *p*[*] |
| Global cognition | | | | | | | | |
| Verbal reasoning | −0.11 (−0.31, 0.09) | 0.368 | −0.45 (−0.66, −0.24) | <0.001 | −0.29 (−0.48, −0.10) | 0.010 | −0.65 (−0.86, −0.44) | <0.001 |
| Non-verbal reasoning | −0.06 (−0.21, 0.09) | 0.429 | −0.27 (−0.42, −0.11) | 0.001 | −0.19 (−0.36, −0.03) | 0.046 | −0.38 (−0.56, −0.19) | <0.001 |
| Academics | | | | | | | | |
| Word reading | −0.11 (−0.22, −0.00) | 0.178 | −0.24 (−0.36, −0.13) | <0.001 | −0.11 (−0.23, 0.00) | 0.068 | −0.36 (−0.49, −0.23) | <0.001 |
| Mathematics | −0.12 (−0.28, 0.05) | 0.348 | −0.34 (−0.52, −0.16) | <0.001 | −0.12 (−0.30, 0.06) | 0.177 | −0.40 (−0.60, −0.20) | <0.001 |
| Attention | | | | | | | | |
| Sustained | −0.05 (−0.26, 0.16) | 0.752 | −0.21 (−0.44, 0.01) | 0.076 | −0.25 (−0.48, −0.03) | 0.052 | −0.41 (−0.66, −0.16) | 0.002 |
| Variability | −0.12 (−0.33, 0.09) | 0.523 | −0.32 (−0.54, −0.09) | 0.017 | −0.21 (−0.41, −0.01) | 0.052 | −0.49 (−0.72, −0.27) | <0.001 |
| Commissions | −0.03 (−0.23, 0.16) | 0.752 | 0.02 (−0.18, 0.23) | 0.835 | −0.12 (−0.31, 0.07) | 0.219 | −0.11 (−0.33, 0.10) | 0.302 |
| Focused attention | −0.12 (−0.31, 0.07) | 0.523 | −0.27 (−0.47, −0.07) | 0.017 | −0.23 (−0.44, −0.02) | 0.052 | −0.41 (−0.64, −0.17) | 0.001 |
| Processing speed | | | | | | | | |
| Visual-motor | −0.11 (−0.28, 0.06) | 0.376 | −0.39 (−0.57, −0.21) | <0.001 | −0.27 (−0.44, −0.09) | 0.005 | −0.47 (−0.66, −0.28) | <0.001 |
| Motor | −0.01 (−0.21, 0.20) | 0.959 | −0.19 (−0.41, 0.03) | 0.096 | −0.25 (−0.46, −0.04) | 0.019 | −0.44 (−0.67, −0.21) | <0.001 |
| Memory | | | | | | | | |
| Span | −0.20 (−0.39, −0.02) | 0.132 | −0.33 (−0.53, −0.14) | 0.003 | −0.21 (−0.38, −0.04) | 0.030 | −0.31 (−0.50, −0.11) | 0.002 |
| New verbal encoding | 0.16 (−0.04, 0.37) | 0.223 | −0.22 (−0.44, −0.01) | 0.055 | −0.28 (−0.49, −0.07) | 0.030 | −0.43 (−0.67, −0.20) | 0.001 |
| Short-term verbal recall | −0.07 (−0.26, 0.13) | 0.495 | −0.23 (−0.44, −0.02) | 0.055 | −0.13 (−0.33, 0.07) | 0.212 | −0.30 (−0.52, −0.08) | 0.009 |
| Long-term verbal recall | 0.12 (−0.08, 0.32) | 0.309 | −0.19 (−0.40, 0.02) | 0.079 | −0.25 (−0.46, −0.03) | 0.030 | −0.44 (−0.68, −0.20) | 0.001 |
| Executive Function | | | | | | | | |
| Perseveration | 0.10 (−0.13, 0.33) | 0.516 | −0.16 (−0.41, 0.08) | 0.180 | 0.06 (−0.19, 0.30) | 0.650 | −0.23 (−0.51, 0.04) | 0.095 |
| Working memory | −0.17 (−0.34, 0.00) | 0.112 | −0.33 (−0.51, −0.15) | 0.002 | −0.22 (−0.38, −0.06) | 0.031 | −0.32 (−0.50, −0.14) | 0.001 |
| Cognitive switching/flexibility | 0.01 (−0.23, 0.26) | 0.908 | −0.35 (−0.60, −0.09) | 0.012 | −0.27 (−0.54, −0.01) | 0.057 | −0.52 (−0.82, −0.22) | 0.001 |
| Verbal fluency | −0.21 (−0.42, 0.00) | 0.112 | −0.38 (−0.60, −0.15) | 0.002 | −0.21 (−0.39, −0.02) | 0.057 | −0.40 (−0.61, −0.19) | 0.001 |

*EAA* epigenetic age acceleration calculated used PCGrimAge epigenetic clock.
[a]Models are adjusted for sex, diagnosis age, BMI, physical activity, smoking, and high-dose IV methotrexate.
[b]Models are adjusted for sex, diagnosis age, BMI, physical activity, smoking, cranial radiation dose, intrathecal methotrexate, high-dose IV methotrexate, high-dose cytarabine, and neurosurgery.
[*]Adjusted for multiple comparisons using the false-discovery rate.

between age at diagnosis and PCPhenoAge EAA (data not shown), however, there were for DunedinPACE, PCGrimAge, and mLTL with visual-motor processing speed as well as between verbal fluency and mLTL (Supplementary Table 14). Interestingly, the association between EAA measures and worse visual-motor processing speed was stronger among those diagnosed ≥10 years of age compared to those diagnosed <10 (e.g., PCGrimAge Age <10 *β* = − 0.35 95% CI: −0.59, −0.10 vs. Age ≥ 10 *β* = −1.04 95% CI: −1.47, −0.60).

**Mediation of the treatment-cognition association by EAA**
Lastly, we conducted causal mediation analyses to examine if associations between treatments for childhood cancer and cognition were mediated by EAA, after adjusting for important confounders such as cancer treatments other than the one selected as the primary exposure, age at diagnosis, sex, BMI, physical activity, and smoking. These models were analyzed for any treatment, EAA, cognitive outcome path that met the following criteria: (1) the treatment has been previously demonstrated to be associated with worse neurocognitive function among survivors of childhood cancer[3], (2) in previously published work[11,43] and our data, the treatment was associated with worse EAA, and (3) in our analyses, EAA was associated with worse neurocognitive functioning. Given the cross-sectional nature of these data, the collinearity between treatments, and the potential for residual confounding, we feel these results should be interpreted cautiously. Therefore, we are only reporting causal mediation results for paths where the total effect, the natural indirect effect, and the percentage mediated were statistically significant (*p* < 0.05, Supplemental Table 15). There were no statistically significant causal mediation paths among

survivors treated without CNS-directed therapies. Among CNS-Treated survivors, DunedinPACE mediated associations between high-dose methotrexate and tests of global cognition (percent mediated [95% CI] = 21.7% [1.7, 41.7]) and attention (19.9% [0.3, 39.4]) accounting for approximately 20% of the total effect. PCGrimAge mediated 28.8% of the association between chest radiation and word reading (28.8% [4.2, 53.4]).

## Discussion
We report associations between epigenetic aging and neurocognitive function in long-term survivors of childhood cancer. Depending on which epigenetic clock was used, EAA was associated with worse attention, processing speed, memory, and executive functions in CNS- and non-CNS-treated survivors. Overall, these data suggest that epigenetic aging and neurocognitive aging may be closely linked and that EAA, specifically defined from PCGrimAge or DunedinPACE, may help identify those at greatest risk for neurocognitive impairment and who would benefit most from intervention. Additionally, given the modifiable nature of these biomarkers, they may hold utility in detecting pre-clinical changes in biological aging in response to interventions designed to improve physiologic and cognitive aging trajectories, such as diet, physical activity, and senolytics[44–46].

When examining either PCPhenoAge or DunedinPACE, two distinct patterns emerged in the CNS-treated and non-CNS-treated groups. The CNS-treated group experienced deficits in memory associated with EAA, while the non-CNS group did not. It may be that the initial neurotoxic effect of therapy makes the CNS-treated group more vulnerable to the effects of continued epigenetic aging on their

**Table 4 | Mean difference (95% CI) in neurocognitive z-score associated with the second or third tertile of DunedinPACE compared to the first tertile stratified by exposure to CNS-directed therapy**

| | Non-CNS-treated[a] n = 663 | | | | CNS-treated[b] n = 750 | | | |
|---|---|---|---|---|---|---|---|---|
| | Tertile 2 vs. 1 | | Tertile 3 vs. 1 | | Tertile 2 vs. 1 | | Tertile 3 vs. 1 | |
| | β (95% CI) | p* | β (95% CI) | p* | β (95% CI) | p* | β (95% CI) | p* |
| **Global cognition** | | | | | | | | |
| Verbal reasoning | -0.15 (-0.36, 0.06) | 0.509 | -0.36 (-0.57, -0.15) | 0.004 | -0.26 (-0.45, -0.07) | 0.016 | -0.41 (-0.62, -0.20) | 0.001 |
| Non-verbal reasoning | -0.05 (-0.20, 0.11) | 0.553 | -0.13 (-0.29, 0.03) | 0.106 | -0.22 (-0.38, -0.06) | 0.016 | -0.32 (-0.50, -0.13) | 0.001 |
| **Academics** | | | | | | | | |
| Word reading | -0.07 (-0.18, 0.05) | 0.509 | -0.17 (-0.28, -0.05) | 0.011 | -0.12 (-0.24, -0.01) | 0.045 | -0.23 (-0.36, -0.10) | 0.001 |
| Mathematics | -0.07 (-0.24, 0.11) | 0.553 | -0.23 (-0.41, -0.05) | 0.018 | -0.18 (-0.35, 0.00) | 0.050 | -0.30 (-0.50, -0.10) | 0.003 |
| **Attention** | | | | | | | | |
| Sustained | 0.18 (-0.04, 0.39) | 0.268 | -0.02 (-0.25, 0.20) | 0.834 | 0.01 (-0.21, 0.23) | 0.899 | -0.33 (-0.58, -0.09) | 0.011 |
| Variability | -0.08 (-0.30, 0.14) | 0.467 | -0.30 (-0.52, -0.07) | 0.038 | -0.17 (-0.36, 0.03) | 0.395 | -0.36 (-0.58, -0.14) | 0.006 |
| Commissions | 0.15 (-0.05, 0.35) | 0.268 | 0.12 (-0.09, 0.33) | 0.439 | 0.08 (-0.11, 0.27) | 0.825 | -0.02 (-0.23, 0.20) | 0.889 |
| Focused attention | 0.13 (-0.07, 0.33) | 0.268 | -0.10 (-0.30, 0.10) | 0.439 | 0.03 (-0.17, 0.24) | 0.899 | -0.35 (-0.58, -0.12) | 0.006 |
| **Processing Speed** | | | | | | | | |
| Visual-motor | 0.13 (-0.05, 0.30) | 0.164 | -0.13 (-0.31, 0.05) | 0.304 | -0.14 (-0.31, 0.04) | 0.244 | -0.36 (-0.55, -0.16) | 0.001 |
| Motor | 0.22 (0.00, 0.43) | 0.099 | -0.11 (-0.34, 0.11) | 0.310 | 0.01 (-0.20, 0.21) | 0.948 | -0.30 (-0.53, -0.07) | 0.012 |
| **Memory** | | | | | | | | |
| Span | 0.06 (-0.14, 0.25) | 0.564 | -0.23 (-0.42, -0.03) | 0.099 | -0.10 (-0.27, 0.07) | 0.252 | -0.12 (-0.31, 0.07) | 0.223 |
| New verbal encoding | 0.15 (-0.06, 0.36) | 0.564 | -0.02 (-0.24, 0.19) | 0.950 | -0.43 (-0.64, -0.23) | <0.001 | -0.36 (-0.59, -0.13) | 0.004 |
| Short-term verbal recall | 0.09 (-0.12, 0.29) | 0.564 | -0.01 (-0.22, 0.20) | 0.950 | -0.32 (-0.52, -0.13) | 0.002 | -0.29 (-0.51, -0.07) | 0.014 |
| Long-term verbal recall | 0.08 (-0.13, 0.29) | 0.564 | 0.02 (-0.20, 0.23) | 0.950 | -0.45 (-0.65, -0.24) | <0.001 | -0.40 (-0.64, -0.17) | 0.003 |
| **Executive Function** | | | | | | | | |
| Perseveration | 0.14 (-0.10, 0.37) | 0.313 | -0.16 (-0.41, 0.08) | 0.417 | -0.11 (-0.35, 0.13) | 0.477 | -0.33 (-0.60, -0.06) | 0.022 |
| Working memory | 0.09 (-0.09, 0.28) | 0.313 | -0.11 (-0.29, 0.08) | 0.417 | -0.09 (-0.25, 0.07) | 0.477 | -0.14 (-0.32, 0.04) | 0.115 |
| Cognitive switching/flexibility | 0.29 (0.03, 0.54) | 0.109 | -0.07 (-0.33, 0.19) | 0.613 | -0.09 (-0.35, 0.17) | 0.477 | -0.58 (-0.88, -0.29) | <0.001 |
| Verbal fluency | 0.16 (-0.06, 0.38) | 0.313 | -0.12 (-0.34, 0.11) | 0.417 | -0.19 (-0.37, -0.00) | 0.184 | -0.28 (-0.49, -0.07) | 0.015 |

EAA epigenetic age acceleration calculated used DunedinPACE epigenetic clock.
[a]Models are adjusted for sex, diagnosis age, BMI, physical activity, smoking, and high-dose IV methotrexate.
[b]Models are adjusted for sex, diagnosis age, BMI, physical activity, smoking, cranial radiation dose, intrathecal methotrexate, high-dose IV methotrexate, high-dose cytarabine, and neurosurgery.
*adjusted for multiple comparisons using the false-discovery rate.

memory-specific cognitive reserve[2]. Interestingly, we also noted associations between EAA and memory span in the HL group, who are treated without CNS-directed therapy. Our previous work demonstrated the HL group to have the highest EAA across diagnoses, possibly because of the strong association between chest radiation and EAA[11]. It is possible that treatment related systemic EAA predisposes HL survivors to chronic health conditions (e.g., cerebrovascular disease) that confer physiologic stress and cause sufficient molecular-level damage such that demand exceeds reserve and results in neurocognitive impairment[2]. Future longitudinal and mechanistic research is warranted in both CNS-treated and HL survivors to understand the potential for a causal relationship between EAA and memory decline. This is critical as these two populations are vulnerable to many aging-related late effects that increase their risk for continued neurocognitive decline that may ultimately manifest as dementia[36].

Similar to our findings, several population-based cohorts report associations between greater EAA based on DNAmPhenoAge and worse cognition, but report no associations with Horvath or Hannum EAA[13,47-49]. The Horvath and Hannum clocks are "first-generation" epigenetic clocks constructed to predict chronological age and hence were suboptimal to predict physiologic aging (e.g., frailty, mortality). In contrast, "second generation" clocks like DNAmPhenoAge and subsequently PCPhenoAge and PCGrimAge were designed to predict a phenotypic aging based on varying sets of laboratory and clinical variables associated with morbidity and mortality[31,33,50]. DunedinPACE was regarded as the "third generation" where DNAm sites were selected to predict the changes of biological aging measurements. DNAmPhenoAge, DunedinPACE, and GrimAge are associated with new onset age-related chronic health conditions[31], which we have previously demonstrated are elevated in survivors of childhood cancer[51] and are associated with an increased risk of neurocognitive impairment[2]. Therefore, these clocks may better represent underlying epigenetic alterations associated with both physiological and neurocognitive aging. Our current data also suggest that PCGrimAge and DunedinPACE may be more sensitive than PCPhenoAge to neurocognitive dysfunction. In non-cancer populations, DunedinPACE and the original GrimAge clocks are highly correlated[33], and have been demonstrated to be better predictors of mortality than DNAmPhenoAge[33,50]. In survivors of childhood cancer, GrimAge appears to be a slightly better predictor of frailty than DNAmPhenoAge[40], and we have demonstrated strong prospective associations between frailty and neurocognitive decline in this group[52]. Therefore, it remains unclear if these markers are surrogates for different pathological processes along the same causal pathway and additional work is needed to understand the functional implications of CpG sites included on each clock. Further, it may be that a new "clock" is needed that is trained to predict this early onset neurocognitive dysfunction among long-term survivors.

Several studies have demonstrated that cancer therapies such as radiation, alkylating agents, corticosteroids, and epipodophyllotoxins can change DNA methylation patterns, which may persist into long-term survivorship where they are compounded by additional environmental insults and biobehavioral changes[8,53]. However, research is limited on EAA and neurocognitive function in patients with cancer, and existing studies predominantly focus on breast cancer[25-28,34]. For example, a small study of women with early-stage breast cancer reported changes in EAA (Horvath's clock) from pre- to post-chemotherapy associated with worse performance on a task of memory at 6 months[28]. Additional CpG level analyses revealed several sites changed after chemotherapy and were associated with worse memory. These CpG sites were mapped to genes associated with neural function and signaling processes[27]. Similarly, a second, larger study of early-stage breast cancer patients demonstrated changes in methylation patterns on genes associated with inflammation from pre- to post-chemotherapy that were subsequently associated with worse self-reported neurocognitive function[25]. The Thinking and Living with

Cancer study observed that patients with breast cancer (age 60+) had older epigenetic age compared to age-matched (within 5 years) non-cancer controls and that older age on extrinsic and GrimAge epigenetic clocks was associated with worse self-reported cognitive function. However, they do not report statistically significant associations between epigenetic age and objective measures of neurocognitive function for DNAmPhenoAge, Horvath, or Hannum clocks[26]. The discrepancy between our studies may be attributable to a longer time since diagnosis, a younger population, and the use of EAA rather than epigenetic age. Importantly, many of these studies demonstrate that epigenetic changes from cancer therapy are maintained up to two years post-treatment[26-28]. These data, combined with our previous work in survivors of childhood cancer[11,12], support the hypothesis that accumulating epigenetic changes from cancer treatment may persist over time and contribute to neurocognitive dysfunction through reduced cognitive reserve and possibly peripheral and neuroinflammatory pathways.

Several studies have demonstrated that chemotherapy and radiation can impart DNA methylation changes at many different CpG sites among patients with breast and gastric cancer[25,27,54-56]. If these changes are sustained over time, it may offer important insights into how cancer treatments might influence biological aging trajectories and increase the risk for neurocognitive impairment. While we do not have access to pre-treatment and immediately post-treatment blood samples from our cohort to directly assess the persistence of methylation changes, we have previously demonstrated that 935 CpG sites with substantial difference in DNA methylation across 538 genes are associated with a history of various childhood cancer treatments (e.g., alkylating agents, radiation) in this same sample of long-term survivors of childhood cancer[57]. Among these CpG sites, only 6 overlap with the CpG sites included on the clocks used in this analysis (Supplementary Table 16). The DNA methylation levels for other clock CpGs may have subtle treatment-related changes (i.e., cannot be detected individually at the epigenome-wide significance level) and contribute cumulatively to the overall observed difference in EAA between exposed and unexposed groups for specific cancer treatment. Further, the cause of cancer-related neurocognitive decline is likely multifactorial, and EAA may only account for a limited proportion of the causal pathway and other mechanisms, such as DNA damage and inflammation should be explored[58,59]. This is highlighted by the lack of mediation by EAA among the survivors treated without CNS-directed therapies. Among CNS-treated survivors, treatment associations with neurocognitive outcomes were only partially mediated by DunedinPACE and PCGrimAge. While this direct overlap between treatment-associated CpGs and epigenetic clock CpGs is limited, the composite DNA methylation alterations in all clock CpGs show a substantial difference reinforcing that cancer treatments influence EAA. Further work is needed to assess whether methylation changes at these loci are durable, directional, and whether they contribute meaningfully to accelerated epigenetic aging over long-term follow-up.

Despite survivors of childhood cancer having significantly shorter telomeres[12], our data do not support an association between mLTL and neurocognitive dysfunction in long-term survivors of childhood cancer. This is consistent with one study of breast cancer survivors, 3-6 years post therapy[60], but in contrast to a study of breast cancer survivors immediately after treatment, which reported that shorter telomeres were associated with worse neurocognitive function in memory, attention, and executive function[61]. Telomere length may be a marker of acute damage and acute neurotoxicity, but it does not represent the protracted aging-related pathways we aim to measure here. In line with this hypothesis, our previous work among childhood cancer survivors demonstrated that while survivors had shorter telomeres than non-cancer controls, this difference did not grow with age and showed a pattern of accentuated aging. In contrast, epigenetic aging showed a pattern of accelerating aging with a steeper slope of annual change of

epigenetic age comparing survivors with non-cancer controls[11,12]. This suggests that telomeres may represent premature aging, or a truncation of physiological reserve after treatment, but do not represent accelerated aging, or continuing decline, and therefore may not be associated with long-term neurocognitive impairment. Consistent with this hypothesis, we saw no overall effect of mLTL on verbal fluency or visual-motor processing speed, but when ALL survivors were stratified by age at diagnosis, among those diagnosed above the age of 10, higher mLTL residual was associated with better performance, but there was no association among those diagnosed under 10.

This study has several strengths, including the large number of long-term survivors of childhood cancer systematically assessed for both epigenetic age and neurocognitive function. We also use five well-validated and commonly used epigenetic clocks. However, these findings are limited by their cross-sectional nature and we cannot draw conclusions on the temporal relationship between epigenetic aging and neurocognition. Future longitudinal studies are needed to elucidate if EAA is truly predictive of neurocognitive decline, if EAA mediates the association between cancer treatment and neurocognitive decline, and if changes in EAA are associated with changes in neurocognitive impairment, especially in the context of health behaviors that may have the potential to reverse EAA (e.g., exercise, smoking cessation). Additionally, longitudinal epigenetic studies are needed in the acute setting to measure changes in EAA from pre- to post-therapy among children and adolescents with cancer. This will inform on the type and magnitude of treatment-related epigenetic changes and how they become persistent in long-term survivorship to induce the associations noted here. Our findings are also likely limited by insufficient sample size and power as several large associations lost statistical significance after adjustment for multiple comparisons. The current analyses were conducted without specific a priori hypotheses about which cognitive measures may be sensitive to epigenetic modification. Therefore, we chose to be conservative and adjust for multiple comparisons to avoid a Type I error. Nonetheless, clinically meaningful effect sizes were noted ranging from 0.2 to 0.3 SD, indicating that future research is needed to replicate these findings, using these data to adequately power those studies to focus on specific subdomains and neurocognitive outcomes of interest. Lastly, our study is limited to survivors of European ancestry and these findings may not be generalizable to other racial/ethnic groups. Research points to disparities in aging by race and ethnicity that may be attributable to a variety of mechanisms, including early life trauma, discrimination, and chronic stress[26,62,63]. Therefore, our findings may be an underestimate of the association between epigenetic aging and cognition in populations not of European ancestry. Current efforts are underway in the SJLIFE cohort to expand the DNA methylation profiling and evaluation of these findings among survivors of non-European ancestry will be carried out when data become available.

In summary, EAA is associated with worse neurocognitive function among long-term survivors of childhood cancer. Specifically, PCGrimAge EAA and DunedinPACE were associated with worse performance across multiple cognitive domains. EAA may identify survivors at risk for accelerated cognitive aging and serve as an efficacy biomarker for neurocognitive interventions in at-risk populations.

## Methods

The SJLIFE protocol, biospecimen banking, and genomic study were approved by the St. Jude institutional review board; participants provided written informed consent. Participants were compensated US $150 per day when they completed their clinical assessment.

### Participants

Survivors treated for pediatric cancer at St. Jude Children's Research Hospital (SJCRH) were enrolled in the St. Jude Lifetime Cohort (SJLIFE); a prospective cohort established to characterize health outcomes among survivors of pediatric cancer[64]. Age-, sex-, and race-matched non-cancer controls were recruited from the same geographic area of the survivors. Eligible survivors were diagnosed between 1962 and 2012 and survived ≥5 years from diagnosis, biospecimens were collected between 2008 and 2016 from the same SJLIFE visit where neurocognitive testing was completed. Assays were performed between 2014 and 2016 for whole genome sequencing (WGS), and between 2018 and 2019 for DNA methylation using available blood samples collected as of March 2016. Survivors of hematopoietic stem cell transplant were excluded due to concerns of genetic/epigenetic profiling representing the donor rather than the survivor. Our analyses were restricted to survivors of European ancestry because mLTL and EAA differ by ancestry[62] and there were too few survivors of non-European ancestry for stratified analyses in the current dataset. Of the 2666 survivors of European ancestry enrolled in SJLIFE as of March 2016, 2252 completed WGS and/or DNA methylation profiling.

Survivors who were non-English speaking, had a genetic or neurodevelopmental syndrome associated with cognitive impairment but unrelated to cancer, or neurologic injury unrelated to cancer treatment (e.g., traumatic brain injury) were excluded. Survivors with CNS tumors were also excluded from the analyses because our prior work suggests the high doses of cranial radiation overpowers genetic influences[65]. Of the 2252 survivors with WGS or DNA methylation profiling data, 1413 (63%) met these additional criteria (Supplementary Fig. 1).

### Procedures

Participants underwent a comprehensive neurocognitive exam concurrent with biospecimen collection. Domains assessed include: attention[66,67], processing speed[68,69], memory[70], and executive function (Supplementary Table 3)[67,69,70]. Raw scores were referenced to national normative data to generate age-adjusted Z-scores. Cumulative chemotherapy, radiation doses and fields, and demographic information were extracted from medical records. BMI and chronic health conditions were ascertained from a clinical exam[71]. Participants self-reported information on biological sex, smoking (current, former, never), physical activity (metabolic equivalent hours per week of vigorous and moderate activity [MET-Hrs]), and education.

As previously described, genome-wide DNA methylation (DNAm) profiling data were generated using the Infinium® MethylationEPIC BeadChip (850 K CpG sites)[11,40]. As described above, we examined five measures of epigenetic aging: PCPhenoAge, PCGrimAge, Horvath and Hannum clocks, and DunedinPACE, all of which have been recently demonstrated to measure aging in survivors of childhood cancer[40]. For each clock (except DunedinPACE), we derived the EAA by calculating the residual for each individual based on the linear regression model of epigenetic age against age at blood draw for DNA sampling.

As previously described, WGS data were generated using the HiSeq X Ten System[72]. TelSeq[73], an extension of a method previously published by our group[74], was used to estimate mLTL from WGS data for the SJLIFE cohort[12]. The TelSeq results correlated highly with Southern blot measurements based on 93 samples from the SJLIFE cohort[12]. mLTL was regressed on chronological age at DNA sampling with additional adjustment for DNA extraction methods as a potential confounder to obtain the residual for subsequent analysis. WGS and methylation data were generated from DNA extracted from the same time point.

### Statistical analysis

We a priori hypothesized that the association between EAA or mLTL may differ in those treated with CNS-directed therapies compared to those treated without them due to previous findings where CNS-treated survivors performed significantly worse on neurocognitive assessments than non-CNS-treated survivors, indicating they may have diminished cognitive reserve as a result of therapy that may

predispose them to aging-related cognitive changes[2]. Therefore, all analyses were stratified by whether or not survivors were exposed to CNS-directed therapy (intrathecal chemotherapy, neurosurgery, or cranial radiation). Demographic and clinical information was compared between survivors treated with and without CNS-directed therapy and non-cancer controls using two-sample t-tests or chi-square tests as appropriate. Among survivors, participants with and without biomarker data were compared in a similar manner to evaluate any potential for selection biases. Linear regression estimated the EAA or mLTL in each group and compared to non-cancer controls adjusted for sex, BMI, smoking, and physical activity.

To increase interpretability, tertiles of each biomarker were generated based on all samples. Separate linear regression models for each neurocognitive outcome estimated the mean difference in age-adjusted neurocognitive z-score associated with the second or third tertile compared to the first. All models were adjusted for sex, age at diagnosis, BMI, smoking, physical activity, and high-dose IV-methotrexate. Among CNS-treated survivors, models were additionally adjusted for cranial radiation dose, intrathecal chemotherapy, and high-dose cytarabine. Because educational attainment is impacted by neurocognitive function after treatment, and because we cannot be sure that education is not on the causal pathway between EAA and neurocognitive function, sensitivity analyses further adjusted models for education and are presented as Supplementary data. Additionally, chronic health conditions were not adjusted for because they are on the causal pathway between EAA and neurocognitive dysfunction and their inclusion would bias our findings[2,11]. Analyses were repeated among subsamples of HL because they experience a disproportionately high amount of EAA[11]. Analyses were also repeated restricted to acute lymphoblastic leukemia (ALL) survivors because they are the most common CNS-treated group and the age at diagnosis extends across the pediatric age range allowing us to evaluate age at diagnosis (<10, 10+) as a modifier. Separate sensitivity analyses among ALL survivors included an interaction term for age and EAA; stratified estimates were generated for any p < 0.05. All hypotheses testing was two-sided. Given the novel nature of this work, we did not have a priori hypotheses about which cognitive domains would be most affected. Therefore, we chose to be conservative and p-values were considered statistically significant at p < 0.05 after adjusting for multiple comparisons using the false discovery rate[41,75].

Lastly, we conducted causal mediation analyses[76,77] to examine if associations between treatments for childhood cancer and cognition were mediated by EAA. These models were analyzed for any treatment, EAA, cognitive outcome path that met the following criteria: (1) the treatment has been previously demonstrated to be associated with worse neurocognitive function among survivors of childhood cancer[3], (2) in previously published work[11,43] and our data, the treatment was associated with worse EAA, and (3) in our analyses, EAA was associated with worse neurocognitive functioning. All causal mediation analyses were adjusted for potential confounders, including age at diagnosis, sex, BMI, physical activity, and smoking. Chronic health conditions and education could not be adjusted for because they are directly impacted by treatment. Models among CNS-treated survivors were additionally adjusted for cranial radiation, intrathecal chemotherapy, high-dose methotrexate, and neurosurgery as appropriate (e.g., when that treatment is not the primary exposure). Given the cross-sectional nature of these data, the collinearity between treatments, and the potential for residual confounding, we feel these results should be interpreted cautiously. Therefore, we are only reporting causal mediation results for paths where the total effect, the natural indirect effect, and the percentage mediated were statistically significant (p < 0.05).

Epigenetic clocks were calculated using pyaging Python package v0.1.22 (available at https://pyaging.readthedocs.io/en/latest/index.html). All subsequent analyses were completed using SAS 9.4 (SAS Institute, Cary, N.C.).

## Reporting summary
Further information on research design is available in the Nature Portfolio Reporting Summary linked to this article.

## Data availability
The publicly available data used in this study are available in the Zenodo database at https://doi.org/10.5281/zenodo.17127194 as well as the St. Jude Cloud at https://stjude.cloud [https://viz.stjude.cloud/cancer-survivorship/visualization/st-jude-survivorship-portal-18]. The DNA methylation data used in this study were generated for a previous study[11], and have been deposited in the NCBI Gene Expression Omnibus database under accession number GSE197674 for survivors [https://www.ncbi.nlm.nih.gov/geo/query/acc.cgi?acc=GSE197674] and accession no. GSE 197676 for controls [https://www.ncbi.nlm.nih.gov/geo/]. The whole genome sequencing data were generated for a previous study[64] and are deposited in the St. Jude Cloud database under accession code SJC-DS-1002. The remaining data are available within the Source data file. Source data are provided with this paper.

## Code availability
The code supporting this study's findings is publicly available at https://doi.org/10.5281/zenodo.17127194.

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

## Acknowledgements

We would like to thank the participants of the SJLIFE cohort for their time and participation in this study. This work was supported by the National Cancer Institute at the National Institutes of Health (grant numbers K99R00CA256356 [Dr. A. Williams], R01CA279520 [Dr. Z. Wang], and U01CA195547 [Drs. M. Hudson and K. Ness]) and the American Lebanese Syrian Associated Charities (ALSAC, all). The content is solely the responsibility of the authors and does not necessarily represent the official views of the National Institutes of Health.

## Author contributions

A.M.W., Q.D., X.M., L.L.R., Z.W., and K.R.K. contributed to study design; A.M.W., N.S.P., M.J.E., N.G., K.P.L., K.K.N., M.M.H., L.L.R., Z.W., and K.R.K. contributed to manuscript writing, reviewing, and editing; Q.D., K.K.N., M.M.H., L.L.R., X.M., Z.W., and K.R.K. prepared materials and collected the data; A.M.W., Q.D., X.M., Z.W., and K.R.K. curated the data; A.M.W., Q.D., and X.M. conducted the statistical analysis; A.M.W., Z.W., and K.R.K. provided project administration and supervised the project and A.M.W. wrote original draft. All authors read and approved the final manuscript.

## Competing interests

The authors declare no competing interests.
