## [Transparent Peer Review file · Nature Communications]

Epigenetic Age Acceleration, Telomere Length, and Neurocognitive Function in Long-Term Survivors of Childhood Cancer

Corresponding Author: Dr AnnaLynn Williams

Version 0:

Reviewer comments:

Reviewer #1

(Remarks to the Author)

The current manuscript represents novel and important findings demonstrating links between epigenetic age and neurocognitive performance in childhood cancer survivors. This data represents one of the first studies to demonstrate these associations in childhood cancer survivors.

The manuscript has a few weaknesses that could be addressed to strengthen the overall impact.

1. Neurocognitive performance is well known to be affected by educational attainment. There is no consideration for this in the analyses and the descriptive data on the participants is very limited (Table 1), including no mention of race/ethnicity, education, comorbidities, BMI, etc. which are all related to epigenetic aging.
2. The current analyses of epigenetic clocks includes older estimates and is not up to date for the field. Resubmission of the epigenetic data to the most up to date software package now located with the clock foundation, will yield further refined and more reliable PhenoAge estimates, called PCPhenoAge, and add important epigenetic clocks including the DunedinPACE and GrimAge.
3. The analyses do not adequately control for potential confounding factors, and secondary models should consider these.
4. Line 170-172 has inaccurate information and the authors should carefully review the paper being referenced.

Reviewer #2

(Remarks to the Author)

This study explores the relationship between biological markers of aging and cognitive functioning in survivors of childhood cancer. The authors found that for survivors treated with CNS-directed therapy, higher epigenetic age acceleration (EAA) was linked to worse performance in attention, verbal encoding, short-term recall, and long-term recall. This suggests that CNS-directed treatment might lead to accelerated biological aging, which could contribute to cognitive declines over time. For survivors not treated with CNS therapy, higher EAA was associated with slower visual-motor processing speed, more perseverations, and reduced cognitive flexibility, indicating that accelerated aging could still impact cognitive function even without CNS-directed treatment. Unlike EAA, mean leukocyte telomere length residual (mLTL) was not found to be associated with cognitive function in this study. The authors conclude that EAA could serve as a biomarker to identify survivors at higher risk of accelerated cognitive aging and to evaluate the effectiveness of interventions aimed at improving cognitive function.

This study has the potential to contribute to our understanding of how epigenetic aging affects neurocognitive outcomes in childhood cancer survivors. However, the authors need to provide more rationale for the analyses present, address several methodological concerns, expand the discussion on the implications of their findings, and provide a more robust justification for their statistical approach.

Major Comments:

1. The hypothesis that cellular aging markers (mLTL and EAA) are associated with neurocognitive impairment in long-term childhood cancer survivors is clear. However, the specific rationale for focusing on certain cognitive domains (memory,

attention, executive function) is not clear. The introduction should better explain why these specific domains are relevant for studying the impact of cellular aging in the context of childhood cancer survivors. Furthermore, the comparison and justification between patients with and without CNS-directed therapy is not well articulated in the introduction.

2. The introduction and discussion could be improved by integrating more recent studies on EAA and neurocognitive outcomes in cancer survivors or similar populations. This would help position the current study within the broader research context.

In addition, this would also support the author's argument on page 3 and 8 that "understanding the relationship between biologic and cognitive aging may inform targeted interventions to mitigate neurocognitive limitations in long-term survivors of childhood cancer" as currently this is not clear.

3. Much of the information needed to interpret and provide rationale for the results is not found until the discussion and methods sections. The manuscript should be rewritten in a format that better adheres to the Nature Communications style – where necessary methodological information and rationale for analyses is summarized in the relevant sections of the results. Much of this information is found in the discussion and methods – but needs to be better situated in the introduction and results to increase interpretability of the results. For example, rationale for comparing the sample with and without biomarker data is not presented. Further, the analytic approach of comparing different tertiles is not described until the methods. The Hannum and Horvath clocks and comparison to them also is presented with little context in the results – as is the concept of epigenetic clock. The rationale for the sub-analyses of HL survivors is not well described.

4. The paper mentions the loss of statistical significance after adjusting for multiple comparisons. A more in-depth discussion of these results, especially in relation to the observed effect sizes, would help in understanding the robustness of these findings despite the loss of statistical significance.

5. The sample itself includes patients treated on protocols from 1962 to 2012. How have treatment protocols changed over this time? It is noted that assays were performed between 2014 and 2016 – but information regarding when the biospecimen samples were collected is not provided. This is important as the epigenetic context can change over time since diagnosis – and this does not appear to be considered. When following diagnosis and treatment that the neurocognitive exam was conducted was not indicated.

6. The study mentions that the cohort is limited to survivors of European ancestry. While the authors acknowledge this as a limitation, they do not provide enough information on how this limitation might affect the generalizability of their results. The paper would benefit from a deeper discussion on how different genetic backgrounds might influence EAA and neurocognitive outcomes.

7. The fact that the CNS-treated group mainly consists of acute lymphoblastic leukemia (ALL) survivors, while the non-CNS-treated group has a significant proportion of Hodgkin lymphoma (HL) survivors, introduces potential confounding variables. The paper should better address how these differences in treatment type and cancer type might affect the observed associations. Furthermore, the paper does not adequately discuss confounding variables that could affect the results, such as socioeconomic status, education level, or other health behaviors, which could influence both epigenetic aging and neurocognitive outcomes.

8. Given the significant differences between the two groups in age at diagnosis and age at treatment, it is important to analyze these variables in relation to the author's neurocognitive outcomes. The timing of treatment during critical periods of brain development can impact a child's cognitive trajectory, with younger patients potentially being more vulnerable to neurocognitive impairment or, alternatively, having greater neuroplasticity to recover. Also, differences in age at treatment might reflect differences in treatment protocols or intensities, which can further influence the degree of cognitive impairment and the biological aging process. Therefore, analyzing age at diagnosis and treatment as covariates is important to understanding the interaction among treatment timing, epigenetic aging, and long-term cognitive outcomes in childhood cancer survivors. Analyzing these associations could reveal whether impairments in neurocognitive function tend to worsen or stabilize as more time passes after treatment. Without considering these factors, the results may overlook important interactions that could inform the study's findings.

9. The relationship between specific cancer therapies, EAA, and neurocognitive outcomes is complex. The authors should consider a more detailed discussion on the mechanisms by which specific treatments might contribute to EAA and subsequent cognitive decline, potentially drawing from preclinical studies or related research in other populations.

Editorial points:

1. HL is not defined upon first use.
2. What is meant by "in a dose dependent manner on line 104/105? Dose of what?"
3. Generally, the paper would benefit from greater attention to editorial detail

Reviewer #3

(Remarks to the Author)

I co-reviewed this manuscript with one of the reviewers who provided the listed reports. This is part of the Nature Communications initiative to facilitate training in peer review and to provide appropriate recognition for Early Career

Researchers who co-review manuscripts.

Reviewer #4

(Remarks to the Author)

The study investigates the relationship between biological aging markers—specifically epigenetic age acceleration (EAA) and mean leukocyte telomere length (mLTL)—and neurocognitive function in childhood cancer survivors. By analyzing three distinct epigenetic clocks (DNAmPhenoAge, Horvath, and Hannum), the authors aim to assess how well these biomarkers predict cognitive impairment in both CNS and non-CNS cancer survivors. Using cross-sectional data from a well-defined survivor cohort of European ancestry, they found that certain EAA measures, particularly DNAmPhenoAge, were significantly associated with cognitive outcomes, suggesting EAA as a potential marker for neurocognitive risk in this population.

Main strengths:

- The study provides a novel perspective on the neurocognitive effects of childhood cancer treatments by incorporating biological aging markers, which have previously been understudied in this context.
- The study benefits from a long follow-up period, tracking participants from their initial cancer diagnoses in childhood through to young adulthood.
- The study's large sample size and rigorous selection criteria improve the reliability and relevance of its findings, and the stratification between CNS and non-CNS cancer survivors offers insight into differing impacts of cancer treatments. A major strength is also having a control group for pertinent comparisons.
- The findings have potential implications for early intervention strategies, as EAA could serve as an actionable biomarker for identifying survivors at risk of cognitive decline.

However, I have some questions and concerns that need to be further addressed by the authors:

- The cross-sectional nature of the study limits causal inferences. It would be useful to assess if methylation mediates the association between cancer treatment vs. controls and neurocognitive function using at least causal mediation analyses.
- Do you have any other variables, such as BMI/weight or presence of other current diseases (diabetes, insulin resistance) that could partly impact neurocognitive function? Adjustment would benefit for adjusting for current BMI or current diseases or medication, since also these variables are known to affect DNA methylation.
- Results in table 2 are only adjusted for sex. It has to be another analysis where further covariates are included to see that the associations are independent of other factors.
- Also in table 2, the comparison is made between cancer survivors and controls. Is there any differences between cancer treatments: Non-CNS vs CNS?
- Were the potential cognitive effects of non-CNS and CNS cancer therapies (such as chemotherapy and radiation) thoroughly accounted for in statistical models? For instance, analyses are adjusted for methotrexate dose but not for radiation nor other chemotherapies.
- Related to my previous question, treatment can affect DNA methylation. There are some studies that have shown methylation sites affected by cancer treatments. Is there any overlap between sites affected by cancer treatment and these sites included in the epigenetic clocks? It would be very interesting since it would mean that treatment can alter methylation, and the methylation in these marks is maintained along this long-follow up period. I guess you do not have blood samples available at cancer diagnosis to actually check methylation maintenance.
- Regarding tables 3 and 4, are there any differences in the cognitive tests of the two survivors' groups separately or even combined compared to controls? It would be useful to see the comparison with the non-cancer individuals.
- Given that telomere length (mLTL) was not significantly associated with neurocognition in the study, could the authors discuss why this might be the case, and whether telomere length remains a relevant biomarker for other aspects of aging in cancer survivors?
- Do the authors believe the biomarkers are specific to neurocognitive decline due to cancer treatment, or could these findings apply to other conditions with neurocognitive impact? Can you discuss more on this topic?
- The study suggests the potential for EAA as a biomarker for intervention. Are there suggestions for how clinicians might use these biomarkers in practice? Could a risk stratification model based on EAA be feasibly implemented? Can you perform further analyses to see whether an optimal cut-point in EAA could be used to stratify those with impaired neurocognitive function and those with not?

Minor comments:

- Variables should be defined in full upon first mention, after which abbreviations may be used consistently.
- Add sample size in all tables.
- If you add data within the manuscript which is not shown in tables, mention "data not shown"
- Check supplementary figure 1 which last number might be wrong
- Can you briefly explain what the TelSeq method is?
- The paper would benefit from including at least one main figure to present the results in a more visual and accessible way.

Version 1:

Reviewer comments:

Reviewer #1

(Remarks to the Author)

Authors have done an excellent job responding to all reviewer points.

Reviewer #2

(Remarks to the Author)

the authors have reasonably address the concerns I raised in my review. I have no further comments.

Reviewer #4

(Remarks to the Author)

Thank you for assessing most of my comments. The manuscript has improved but there are still two analyses that I would like to see:

- Causal mediation analyses to check whether methylation mediates the association between cancer treatment vs. controls and neurocognitive function.
- Any of the methylation sites included in clocks is affected by cancer treatment in previous studies? Perform an overlap with findings from previous studies in the literature

Version 2:

Reviewer comments:

Reviewer #4

(Remarks to the Author)

The authors have adequately addressed the concerns raised in my previous review. I have no further comments.

We would like to thank the reviewers for their thoughtful comments and suggestions. We have included a point-point response below (in italics) as well as used track changes throughout the manuscript. Because all of the tables have been replaced, in the manuscript, and supplement, we did not use track changes on tables and figures. We believe that these changes greatly improved the manuscript and made our findings more robust.

Reviewer #1 (Remarks to the Author):

The current manuscript represents novel and important findings demonstrating links between epigenetic age and neurocognitive performance in childhood cancer survivors. This data represents one of the first studies to demonstrate these associations in childhood cancer survivors.

Response: We would like to thank the reviewer for their supportive words regarding the importance of this work.

The manuscript has a few weaknesses that could be addressed to strengthen the overall impact.

1. Neurocognitive performance is well known to be affected by educational attainment. There is no consideration for this in the analyses and the descriptive data on the participants is very limited (Table 1), including no mention of race/ethnicity, education, comorbidities, BMI, etc. which are all related to epigenetic aging.

Response: We agree with the reviewer, these are important covariables to consider when examining either epigenetic aging or cognition and have added the descriptives of these variables to Table 1, except for race and ethnicity. As described in our methods, the entire sample is of European ancestry and also identifies as non-Hispanic White. We have also further adjusted all primary analyses for body mass index (BMI), smoking, and physical activity.

We agree that education is closely related to neurocognitive test performance. However, childhood cancer survivors often develop neurocognitive problems before they reach their final educational attainment, resulting in significantly lower educational attainment than their siblings and peers. Making education level downstream of their neurocognitive functioning. We also hypothesize that their cancer treatment imparts epigenetic changes that may further influence their educational attainment (through mediators like chronic health conditions), and because we cannot be sure when the epigenetic changes or the neurocognitive function we are measuring began, it is possible that education is on the causal pathway between EAA and neurocognitive function making it inappropriate to adjust for. Therefore, we chose to present sensitivity analyses adjusted for education in the supplement. After doing so, some estimates were attenuated and lost statistical significance.

For similar reasons, we cannot adjust for chronic health conditions. Our prior work (PMID: 32970815) demonstrated that accelerated epigenetic aging was associated with an increased risk of incident chronic health conditions, and we have also demonstrated that chronic health conditions are associated with an increased risk of neurocognitive impairment (PMID: 33274777). Therefore, we feel that comorbidities are on the causal pathway between EAA and neurocognitive function, and inclusion would bias our findings.

Our primary results are now further adjusted for BMI, smoking, and physical activity as recommended by all reviewers.

2. The current analyses of epigenetic clocks includes older estimates and is not up to date for the field. Resubmission of the epigenetic data to the most up to date software package now located with the clock foundation, will yield further refined and more reliable PhenoAge estimates, called PCPhenoAge, and add important epigenetic clocks including the DunedinPACE and GrimAge.

Response: We are grateful to the reviewer for this insightful suggestion as it has resulted in a more robust analysis and additional new findings worthy of highlighting. We have updated our results throughout the paper in two ways; 1) all DNAmPhenoAge analyses have been replaced with PCPhenoAge as suggested (Figure 1, Table 2 and Supplemental Tables 2, 4, 10 and 11) and 2) DunedinPACE and PCGrimAge analyses have been added (Figure 1, Tables 3 and 4 as well as Supplemental Tables 2, 5, 6, 11, 13, and 14). Interestingly, the associations between PCGrimAge or DunedinPACE and cognitive function were stronger and consistent across neurocognitive domains.

We have revised the discussion on page 10 to highlight these findings:

“Similar to our findings, several population-based cohorts report associations between greater EAA based on DNAmPhenoAge and worse cognition, but report no associations with Horvath or Hannum EAA.[13, 46-48] The Horvath and Hannum clocks are “first generation” epigenetic clocks constructed to predict chronological age and hence were suboptimal to predict physiologic aging (e.g., frailty, mortality). In contrast, “second generation” clocks like DNAmPhenoAge and subsequently PCPhenoAge and PCGrimAge were designed to predict a phenotypic aging based on varying sets of laboratory and clinical variables associated with morbidity and mortality.[31, 33, 49] DunedinPACE was regarded as the “third generation” where DNAm sites were selected to predict the changes of biological aging measurements. DNAmPhenoAge, DunedinPACE, and GrimAge are associated with new onset age-related chronic health conditions[31], which we have previously demonstrated are elevated in survivors of childhood cancer[50] and are associated with an increased risk of neurocognitive impairment.[2] Therefore, these clocks may better represent underlying epigenetic alterations associated with both physiological and neurocognitive aging. Our current data also suggest that PCGrimAge and DunedinPACE may be more sensitive than PCPhenoAge to neurocognitive dysfunction. In non-cancer populations, DunedinPACE and the original GrimAge clocks are highly correlated[33], and have been demonstrated to be better predictors of mortality than DNAmPhenoAge.[33, 49] In survivors of childhood cancer, GrimAge appears to be a slightly better predictor of frailty than DNAmPhenoAge[40], and we have demonstrated strong prospective associations between frailty and neurocognitive decline in this group.[51] Therefore, it remains unclear if these markers are surrogates for different pathological processes along the same causal pathway and additional work is needed to understand the functional implications of CpG sites included on each clock. Further, it may be that a new “clock” is needed that is trained to predict this early onset neurocognitive dysfunction among long-term survivors.”

3. The analyses do not adequately control for potential confounding factors, and secondary models should consider these.

Response: We agree with this reviewer and others that important confounders were not accounted for in our original analyses. As mentioned in our response above, all models are now further adjusted for BMI, smoking and physical activity, and sensitivity analyses adjusted for education are presented in the Supplement.

4. Line 170-172 has inaccurate information and the authors should carefully review the paper being referenced.

Response: We have rephrased the results referenced as follows:

Page 11, "The Thinking and Living with Cancer study observed that patients with breast cancer (age 60+) had older epigenetic age compared to age-matched (within 5-years) non-cancer controls and that older age on extrinsic and GrimAge epigenetic clocks was associated with worse self-reported cognitive function. However, they do not report statistically significant associations between epigenetic age and objective measures of neurocognitive function for DNAmPhenoAge, Horvath, or Hannum clocks. [26] The discrepancy between our studies may be attributable to a longer-time since diagnosis, younger population, and the use of EAA rather than epigenetic age."

Reviewer #2 (Remarks to the Author):

This study explores the relationship between biological markers of aging and cognitive functioning in survivors of childhood cancer. The authors found that for survivors treated with CNS-directed therapy, higher epigenetic age acceleration (EAA) was linked to worse performance in attention, verbal encoding, short-term recall, and long-term recall. This suggests that CNS-directed treatment might lead to accelerated biological aging, which could contribute to cognitive declines over time. For survivors not treated with CNS therapy, higher EAA was associated with slower visual-motor processing speed, more perseverations, and reduced cognitive flexibility, indicating that accelerated aging could still impact cognitive function even without CNS-directed treatment. Unlike EAA, mean leukocyte telomere length residual (mLTL) was not found to be associated with cognitive function in this study. The authors conclude that EAA could serve as a biomarker to identify survivors at higher risk of accelerated cognitive aging and to evaluate the effectiveness of interventions aimed at improving cognitive function.

This study has the potential to contribute to our understanding of how epigenetic aging affects neurocognitive outcomes in childhood cancer survivors. However, the authors need to provide more rationale for the analyses present, address several methodological concerns, expand the discussion on the implications of their findings, and provide a more robust justification for their statistical approach.

Response: We would like to thank the reviewer for their positive comments and have tried to address each of their comments below.

Major Comments:

1. The hypothesis that cellular aging markers (mLTL and EAA) are associated with neurocognitive impairment in long-term childhood cancer survivors is clear. However, the specific rationale for focusing on certain cognitive domains (memory, attention, executive

function) is not clear. The introduction should better explain why these specific domains are relevant for studying the impact of cellular aging in the context of childhood cancer survivors. Furthermore, the comparison and justification between patients with and without CNS-directed therapy is not well articulated in the introduction.

Response: We would like to thank the reviewer for bringing this to our attention. We revised the second paragraph of the introduction to indicate that the cognitive domains of memory, attention, and executive functions were of particular interest because of their known dysfunction in aging-related cognitive disorders such as dementia. The manuscript has now been revised as follows:

Page 3: "These well-established aging-related biomarkers have previously been associated with neurocognitive impairment and decline in older non-cancer populations, particularly in cognitive domains related to aging and dementia such as memory, attention, and executive function, but these associations have yet to be examined among younger survivors of childhood cancer.[13-24]

This is accompanied by a sentence in the last paragraph of the introduction on page 4 that now reads: "We hypothesized that these markers of cellular aging would be associated with worse neurocognitive function, especially in the domains of memory, attention and executive function which are commonly impaired in the aging non-cancer population."

We have also added a sentence to the last paragraph of the introduction to describe the rationale of stratifying by the receipt of CNS directed therapy as follows:

Page 4: "We aimed to examine these associations separately among survivors who were and were not treated with therapies directed to the central nervous system as the direct damage from CNS directed therapies may increase vulnerability to aging-related changes in cognitive function due to diminished cognitive reserve.[2, 38]"

This information has also been included in the beginning of the results:

Page 5: "Analyses were stratified by receipt of CNS-directed therapy during treatment for childhood cancer as previous research has demonstrated this is associated with increased vulnerability to aging-related changes due to diminished cognitive reserve.[2, 39]"

2. The introduction and discussion could be improved by integrating more recent studies on EAA and neurocognitive outcomes in cancer survivors or similar populations. This would help position the current study within the broader research context. In addition, this would also support the author's argument on page 3 and 8 that "understanding the relationship between biologic and cognitive aging may inform targeted interventions to mitigate neurocognitive limitations in long-term survivors of childhood cancer" as currently this is not clear.

Response: In addition to changes to the discussion, we have added the following paragraph to the introduction as follows:

Pages 3-4: "Several studies have demonstrated that chemotherapy and radiation can impart significant DNA methylation changes in the context of breast cancer and that these changes are associated with functional and cognitive impairments.[25-27] Importantly, epigenetic changes

were sustained two years post-therapy, suggesting that epigenetic alterations from cancer treatment persist into long-term survivorship.[27, 28] Given the vast number of CpG sites affected, epigenetic clocks may serve as a way to synthesize these data into one composite measure of biological age. At present, several epigenetic clocks have been created, with the first generation of clocks trained to predict chronological age (e.g. Horvath[29], Hannum[30]), the second generation of clocks trained to predict morbidity and mortality (e.g. Levine's PCPhenoAge[31, 32], PCGrimAge[32]), and the third generation focused on predicting the pace of aging (e.g. DunedinPACE[33]). Sehl and colleagues have reported significant increases in various epigenetic clocks from pre- to post-chemotherapy and radiation treatment for breast cancer.[34] Similarly, findings from the Thinking and Living with Cancer study suggest increases in epigenetic age are sustained 2-3 years post-breast cancer treatment and are associated with functional and cognitive impairments.[26] These findings highlight that epigenetic modifications from cancer treatment may extend well into survivorship, consistent with our cross-sectional work in long-term survivors of childhood cancer that suggests they experience significant epigenetic age acceleration 10 or more years from treatment.[11, 12, 35] However, the association between epigenetic modifications and long-term neurocognitive impairment has yet to be examined among survivors of childhood cancer who may be experiencing premature cognitive aging.[3, 36, 37] “

We have also added the following to the beginning of the discussion as follows:

Page 9: “Overall, these data suggest that epigenetic aging and neurocognitive aging may be closely linked and that EAA, specifically defined from PCGrimAge or DunedinPACE, may help identify those at greatest risk for neurocognitive impairment and who would benefit most from intervention. Additionally, given the modifiable nature of these biomarkers, they may hold utility in detecting pre-clinical changes in biological aging in response to interventions designed to improve physiologic and cognitive aging trajectories such as diet, physical activity, and senolytics.[43-45]”

We have altered the last sentence of the introduction as follows:

Page 4: “Given that epigenetic changes are modifiable, understanding the relationship between biologic and cognitive aging may inform targeted interventions to mitigate neurocognitive limitations in long-term survivors of childhood cancer.”

3. Much of the information needed to interpret and provide rationale for the results is not found until the discussion and methods sections. The manuscript should be rewritten in a format that better adheres to the Nature Communications style – where necessary methodological information and rationale for analyses is summarized in the relevant sections of the results. Much of this information is found in the discussion and methods – but needs to be better situated in the introduction and results to increase interpretability of the results. For example, rationale for comparing the sample with and without biomarker data is not presented. Further, the analytic approach of comparing different tertiles is not described until the methods. The Hannum and Horvath clocks and comparison to them also is presented with little context in the results – as is the concept of epigenetic clock. The rationale for the sub-analyses of HL survivors is not well described.

Response: We have made various edits throughout the paper to adhere to the Nature Communications Style. Specifically, we have added the following statements to the Results section to clarify the reviewer's questions:

Page 5: "We first compared survivors in our analytic sample to the remaining survivors in SJLIFE to examine any potential for selection bias."

Page 6: "To increase interpretability, tertiles of each biomarker were generated based on all samples for use in linear regression models that estimated mean differences in age-adjusted neurocognitive z-scores across tertiles of EAA/mLTL, adjusting for sex, age at diagnosis, body mass index (BMI), smoking, physical activity, and treatments associated with worse neurocognitive impairment."

Page 5: "We examined the association between neurocognitive function and biologic aging among long-term survivors of childhood cancer using mLTL and several epigenetic clocks as it is currently unclear which epigenetic clock may best predict neurocognitive dysfunction in this population. PCPhenoAge was initially chosen as our primary outcome because it's original derivation, DNAmPhenoAge, outperforms other epigenetic clocks in terms of prediction of all-cause mortality, and age-related health outcomes in the general population and has been recently used to demonstrate accelerated aging in childhood cancer survivors.[11, 29, 32, 35, 40] We examined four other epigenetic clocks (PCGrimAge[32], DunedinPACE[33], Horvath[29] Hannum[30]) that have also recently been demonstrated to characterize accelerated aging in childhood cancer survivors[40]. Notably, DunedinPACE was developed among a sample of young adults and may align better to our sample of young adult survivors of childhood cancer. We used the principal components (PC) version of DNAmPhenoage (and GrimAge) to reduce noise due to technical variability of individual DNA methylation sites.[32] For each clock (except DunedinPACE), we derived the EAA by calculating the residual for each individual based on the linear regression model of epigenetic age against age at blood draw for DNA sampling."

Page 7: "We conducted sensitivity analyses among survivors of HL, because they experience a disproportionately high amount of epigenetic age acceleration compared to other diagnoses (Supplemental Table 10).[11]"

4. The paper mentions the loss of statistical significance after adjusting for multiple comparisons. A more in-depth discussion of these results, especially in relation to the observed effect sizes, would help in understanding the robustness of these findings despite the loss of statistical significance.

Response: We present one of the first analyses to examine the relationship between epigenetic aging and cognitive function in survivors of childhood cancer. We considered these analyses to be exploratory, with no specific a priori hypotheses, therefore, to be more conservative, we adjusted all of our analyses for multiple comparisons in order to avoid a type I error. As the reviewer suggests, we still see clinically meaningful effect sizes (e.g. 0.2 to 0.3 SD difference in age adjusted Z-scores associated with each tertile of EAA), indicating these findings may in fact be robust and should be investigated further. We hope that future research can replicate these findings, focusing on specific cognitive domains and informative epigenetic clocks, and use these data to adequately power analyses to solidify these effects.

We have added the following to the results on page 6: “Because this is the first study to examine epigenetic aging and neurocognitive function among long-term survivors of childhood cancer, we did not have a priori hypotheses about which domains and which neuropsychological assessments would be most affected. Therefore, we chose to be conservative and adjust p-values for multiple comparisons using the false discovery rate.[41]”

We have added the following to the Discussion on pages 12 and 13: “Our findings are also likely limited by insufficient sample size and power as several large associations lost statistical significance after adjustment for multiple comparisons. The current analyses are novel and were conducted without specific a priori hypotheses about which cognitive measures may be sensitive to epigenetic modification. Therefore, we chose to be conservative and adjust for multiple comparisons to avoid a Type I error. Nonetheless, clinically meaningful effect sizes were noted ranging from 0.2 to 0.3 SD, indicating that future research is needed to replicate these findings, using these data to adequately power those studies to focus on specific subdomains and neurocognitive outcomes of interest.”

5. The sample itself includes patients treated on protocols from 1962 to 2012. How have treatment protocols changed over this time? It is noted that assays were performed between 2014 and 2016 – but information regarding when the biospecimen samples were collected is not provided. This is important as the epigenetic context can change over time since diagnosis – and this does not appear to be considered. When following diagnosis and treatment that the neurocognitive exam was conducted was not indicated.

Response: Our sample does include long-term survivors originally diagnosed and treated from 1962 to 2012 and there have been significant treatment changes over time. However, the “backbone” of treatment regimens has largely remained the same, with changes to treatment dosing and order being more common. The samples used in this study were collected in long-term survivorship, from 2008 to 2016, from the same SJLIFE visit the neurocognitive battery was completed. Therefore, the primary aim of this study is to examine the association between biologic aging and neurocognitive aging in the very long-term setting. The initial comparison of survivors and controls is provided for context, to demonstrate that survivors experience significantly increased epigenetic aging, even in our nested subsample of the large SJLIFE cohort. The original paper by Qin et al details the treatment relationships between epigenetic aging and explores these issues including associations between specific treatment exposure and epigenetic age acceleration (PMID: 32970815).

We have clarified in the methods on page 13: “Eligible survivors were diagnosed between 1962 and 2012 and survived ≥ 5 years from diagnosis, biospecimens were collected between 2008 and 2016 from the same SJLIFE visit where neurocognitive testing was completed. Assays were performed between 2014 and 2016 for whole genome sequencing (WGS), and between 2018 and 2019 for DNA methylation using available blood samples collected as of March 2016.”

6. The study mentions that the cohort is limited to survivors of European ancestry. While the authors acknowledge this as a limitation, they do not provide enough information on how this limitation might affect the generalizability of their results. The paper would benefit from a deeper discussion on how different genetic backgrounds might influence EAA and neurocognitive outcomes.

Response: We have expanded the discussion to include our hypothesis that these findings are not representative of survivors of non-European ancestry and that the association we report is likely an underestimate of the association we would see among those of other ancestry. On Page 13, the discussion now reads: "Research points to disparities in aging by race and ethnicity that may be attributable to a variety of mechanisms including early life trauma, discrimination, and chronic stress.[26, 55] Therefore, our findings may be an underestimate of the association between epigenetic aging and cognition in populations not of European ancestry. Current efforts are underway in the SJLIFE cohort to expand the DNA methylation profiling and evaluation of these findings among survivors of non-European ancestry will be carried out when data become available."

7. The fact that the CNS-treated group mainly consists of acute lymphoblastic leukemia (ALL) survivors, while the non-CNS-treated group has a significant proportion of Hodgkin lymphoma (HL) survivors, introduces potential confounding variables. The paper should better address how these differences in treatment type and cancer type might affect the observed associations. Furthermore, the paper does not adequately discuss confounding variables that could affect the results, such as socioeconomic status, education level, or other health behaviors, which could influence both epigenetic aging and neurocognitive outcomes.

Response: The reviewer is correct that the majority of the CNS-treated group are ALL survivors (77%) followed by NHL (14%). While the non-CNS group is primarily composed of HL, this only accounts for 32% of the group. Nonetheless, we conducted sensitivity analyses within these primary diagnoses to examine the impact of the issue the authors raise, wondering if there was confounding by something specific to their host biology that predisposed them to both epigenetic aging and their cancer. Additionally, we felt it was important to examine HL separately as our previous work has reported they experience the greatest levels of EAA compared to other diagnosis groups.

Not unexpectedly, results between ALL survivors and the entire CNS treated group were somewhat similar. Among HL survivors, new associations specific to memory span with each measure of EAA revealed itself. The reasons for this remains unclear and as the reviewer points out there may be some yet unaccounted for residual confounding. Nonetheless, models are adjusted for all known treatment risk factors for cognitive impairment. Future research is needed with longitudinal data pre and post treatment to understand these nuances.

On Page 12, the Discussion now reads: "Future longitudinal studies are needed to elucidate if EAA is truly predictive of neurocognitive decline, if EAA mediates the association between cancer treatment and neurocognitive decline, and if changes in EAA are associated with changes in neurocognitive impairment, especially in the context of health behaviors that may have the potential to reverse EAA (e.g. exercise, smoking cessation). Additionally, longitudinal epigenetic studies are needed in the acute setting to measure changes in EAA from pre- to post-therapy among children and adolescents with cancer. This will inform on the type and magnitude of treatment-related epigenetic changes and how they become persistent in long-term survivorship to induce the associations noted here."

Lastly, we have added information on potential confounding variables such as education, BMI, smoking, and physical activity to Table 1. We have also adjusted all primary analyses for sex, age at diagnosis, BMI, smoking, physical activity and all treatments recognized to increase the risk of neurocognitive impairment. As noted in our response to Reviewer 1, comorbidities are

likely on the causal pathway between EAA and cognitive function and adjustment would bias our findings. Similarly, it is unclear if education may also be on the causal pathway, therefore sensitivity analyses adjusting for education are now presented in the supplement.

8. Given the significant differences between the two groups in age at diagnosis and age at treatment, it is important to analyze these variables in relation to the author's neurocognitive outcomes. The timing of treatment during critical periods of brain development can impact a child's cognitive trajectory, with younger patients potentially being more vulnerable to neurocognitive impairment or, alternatively, having greater neuroplasticity to recover. Also, differences in age at treatment might reflect differences in treatment protocols or intensities, which can further influence the degree of cognitive impairment and the biological aging process. Therefore, analyzing age at diagnosis and treatment as covariates is important to understanding the interaction among treatment timing, epigenetic aging, and long-term cognitive outcomes in childhood cancer survivors. Analyzing these associations could reveal whether impairments in neurocognitive function tend to worsen or stabilize as more time passes after treatment. Without considering these factors, the results may overlook important interactions that could inform the study's findings.

*Response: Age at diagnosis and age at treatment are very similar in this cohort, as all survivors are treated upon diagnosis. We also have adjusted all analyses for age at diagnosis. However, we agree with the reviewer that age at treatment often has a large impact on subsequent cognitive function. This is particularly true for cranial radiation among patients with ALL, where cranial radiation prior to the age of 10 is associated with worse subsequent cognitive functioning. To address the reviewer's question about a potential effect modification by age at diagnosis, we added an age*EAA or age*mLTL interaction term to our ALL-specific model. We chose to restrict this analysis to ALL because they are the only diagnosis group with sufficient numbers spanning the age continuum and because the heterogeneity in treatment may mask age effects if examined in the entire cohort. There was a significant interaction with age at diagnosis for DunedinPACE, PCGrimAge, and mLTL for performance on a test of visual-motor processing speed (Supplemental Table 14).*

In the results on page 9, we have added: "Previous research suggests that age at diagnosis and treatment can impact survivor's vulnerability to short- and long-term cognitive impairment.[42] Therefore, we examined if there was a statistical interaction between age at diagnosis and EAA. There were no statistically significant interactions between age at diagnosis and PCPhenoAge EAA (data not shown), however, there were for DunedinPACE, PCGrimAge, and mLTL with visual-motor processing speed as well as between verbal fluency and mLTL (Supplemental Table 14). Interestingly, the association between EAA measures and worse visual-motor processing speed was stronger among those diagnosed ≥ 10 years of age compared to those diagnosed < 10 (e.g. PCGrimAge Age < 10 $\beta = -0.35$ 95%CI -0.59, -0.10 vs. Age ≥ 10 $\beta = -1.04$ 95%CI -1.47, -0.60)."

In the methods on page 16, we have added: "Analyses were also repeated restricted to acute lymphoblastic leukemia (ALL) survivors because they are the most common CNS-treated group and the age at diagnosis extends across the pediatric age range allowing us to evaluate age at diagnosis (< 10 , $10+$) as a modifier. Separate sensitivity analyses among ALL survivors included an interaction term for age and EAA, stratified estimates were generated for any $p < 0.05$. All hypotheses testing was 2-sided."

9. The relationship between specific cancer therapies, EAA, and neurocognitive outcomes is complex. The authors should consider a more detailed discussion on the mechanisms by which specific treatments might contribute to EAA and subsequent cognitive decline, potentially drawing from preclinical studies or related research in other populations.

Response: We agree with reviewers that the relationship between various cancer therapies, epigenetic changes, and changes in neurocognitive outcomes is very complex. However, this study is done in the context of long-term survivorship, 20+ years from treatment, and the initial molecular damage from treatment may be compounded over time by additional environmental and biobehavioral insults. Therefore, we provide a broad overview for the basis of the initial molecular insult at treatment, during development, and then frame our discussion in the setting of long-term survivorship and premature aging.

On pages 11 and 12 of the discussion: "Several studies have demonstrated that cancer therapies such as radiation, alkylating agents, corticosteroids, and epipodophyllotoxins can change DNA methylation patterns which may persist into long-term survivorship where they are compounded by additional environmental insults and biobehavioral changes.".... "Additionally, longitudinal epigenetic studies are needed in the acute setting to measure changes in EAA from pre- to post-therapy among children and adolescents with cancer. This will inform on the type and magnitude of treatment-related epigenetic changes and how they become persistent in long-term survivorship to induce the associations noted here."

On pages 3 and 4 of the introduction we have added the following: "Several studies have demonstrated that chemotherapy and radiation can impart significant DNA methylation changes in the context of breast cancer and that these changes are associated with functional and cognitive impairments.[25-27] Importantly, epigenetic changes were sustained two years post-therapy, suggesting that epigenetic alterations from cancer treatment persist into long-term survivorship.[27, 28] Given the vast number of CpG sites affected, epigenetic clocks may serve as a way to synthesize these data into one composite measure of biological age. At present, several epigenetic clocks have been created, with the first generation of clocks trained to predict chronological age (e.g. Horvath[29], Hannum[30]), the second generation of clocks trained to predict morbidity and mortality (e.g. Levine's PCPhenoAge[31, 32], PCGrimAge[32]), and the third generation focused on predicting the pace of aging (e.g. DunedinPACE[33]). Sehl and colleagues have reported significant increases in various epigenetic clocks from pre- to post-chemotherapy and radiation treatment for breast cancer.[34] Similarly, findings from the Thinking and Living with Cancer study suggest increases in epigenetic age are sustained 2-3 years post-breast cancer treatment and are associated with functional and cognitive impairments.[26] These findings highlight that epigenetic modifications from cancer treatment may extend well into survivorship, consistent with our cross-sectional work in long-term survivors of childhood cancer that suggests they experience significant epigenetic age acceleration 10 or more years from treatment.[11, 12, 35] However, the association between epigenetic modifications and long-term neurocognitive impairment has yet to be examined among survivors of childhood cancer who may be experiencing premature cognitive aging.[3, 36, 37]."

Editorial points:

1. HL is not defined upon first use.

Response: Thank you for pointing out this error, we have corrected it.

2. What is meant by “in a dose dependent manner on line 104/105? Dose of what?”

Response: This refers to the fact that the effect sizes increase from tertile 2 to tertile 3. Indicating that as epigenetic age acceleration tertile increases, their performance worsens. However, after converting our analysis to the PCPhenoAge, this was no longer true and the sentence has been revised as follows: “ Among HL, higher PCPhenoAge EAA was associated with worse memory span, although not statistically significant after adjustment for multiple comparisons ($\beta=-0.58$ 95%CI -1.05, -0.11; $p=0.063$ Supplemental Table 10) as was DunedinPACE ($\beta=-0.47$ 95%CI -0.87, -0.08; $p=0.075$, Supplemental Table 11) and PCGrimAge ($\beta=-0.66$ 95%CI -1.01, -0.30, $p=0.001$).”

We did observe similar patterns when using PCGrimAge among CNS-treated survivors and have included the following interpretation on page 8 of our results: “When PCGrimAge was used as the measure of EAA, statistically significant associations were noted for each cognitive domain across both tertile 2 and tertile 3, with effect sizes increasing as tertile of EAA increased (Table 3). For example, those in the second tertile of PCGrimAge EAA performed a quarter of a SD worse, on average, on a test of long-term recall compared to tertile 1, while those in tertile 3 performed 0.44 SD worse, on average, compared to tertile 1 (tertile 2 $\beta=-0.25$ 95%CI -0.46, -0.03; $p=0.030$, tertile 3 $\beta=-0.44$ 95%CI -0.68, -0.20; $p=0.001$). Similar associations across tertiles were noted for tests of sustained attention, attention variability, visual-motor and motor processing speed, memory span, and working memory (p 's <0.05).

3. Generally, the paper would benefit from greater attention to editorial detail

Response: We have thoroughly revised the paper.

Reviewer #3 (Remarks to the Author):

Reviewer #4 (Remarks to the Author):

The study investigates the relationship between biological aging markers—specifically epigenetic age acceleration (EAA) and mean leukocyte telomere length (mLTL)—and neurocognitive function in childhood cancer survivors. By analyzing three distinct epigenetic

clocks (DNAmPhenoAge, Horvath, and Hannum), the authors aim to assess how well these biomarkers predict cognitive impairment in both CNS and non-CNS cancer survivors. Using cross-sectional data from a well-defined survivor cohort of European ancestry, they found that certain EAA measures, particularly DNAmPhenoAge, were significantly associated with cognitive outcomes, suggesting EAA as a potential marker for neurocognitive risk in this population.

Main strengths:

- The study provides a novel perspective on the neurocognitive effects of childhood cancer treatments by incorporating biological aging markers, which have previously been understudied in this context.
- The study benefits from a long follow-up period, tracking participants from their initial cancer diagnoses in childhood through to young adulthood.
- The study's large sample size and rigorous selection criteria improve the reliability and relevance of its findings, and the stratification between CNS and non-CNS cancer survivors offers insight into differing impacts of cancer treatments. A major strength is also having a control group for pertinent comparisons.
- The findings have potential implications for early intervention strategies, as EAA could serve as an actionable biomarker for identifying survivors at risk of cognitive decline.

Response: We would like to thank the reviewer for highlighting the strengths and importance of this research.

However, I have some questions and concerns that need to be further addressed by the authors:

- The cross-sectional nature of the study limits causal inferences. It would be useful to assess if methylation mediates the association between cancer treatment vs. controls and neurocognitive function using at least causal mediation analyses.

Response: We agree with the reviewer, the cross-sectional nature of these data do limit our ability to truly inform on the potential causal mechanisms at play and that a mediation analysis could shed some light on this topic. We are in the process of assaying serial samples from these survivors that will yield both prospective epigenetic data and prospective cognitive data. We feel that mediation analyses with longitudinal data will be able to more robustly address this question and plan to do this in our future work. We have added this to the discussion on page 12: "Future longitudinal studies are needed to elucidate if EAA is truly predictive of neurocognitive decline, if EAA mediates the association between cancer treatment and neurocognitive decline, and if changes in EAA are associated with changes in neurocognitive impairment, especially in the context of health behaviors that may have the potential to reverse EAA (e.g. exercise, smoking cessation)."

- Do you have any other variables, such as BMI/weight or presence of other current diseases (diabetes, insulin resistance) that could partly impact neurocognitive function? Adjustment would benefit for adjusting for current BMI or current diseases or medication, since also these variables are known to affect DNA methylation.

Response: We agree with the reviewers, and have presented data on BMI, chronic health conditions, physical activity, education, and smoking in Table 1. We now present as our primary analyses models adjusted for sex, age at diagnosis, BMI, physical activity, smoking and treatment associated with neurocognitive impairment. As detailed in our response to Reviewer 1, comorbidities are likely on the causal pathway between EAA and cognitive function and adjustment would bias our findings. Similarly, it is unclear if education may also be on the causal pathway, therefore sensitivity analyses adjusting for education are now presented in the supplement.

- Results in table 2 are only adjusted for sex. It has to be another analysis where further covariates are included to see that the associations are independent of other factors.

Response: We agree, we have now adjusted these analyses for the potential confounders noted above. Additionally, this table has been turned into a figure and full results are presented in Supplemental Table 2

- Also in table 2, the comparison is made between cancer survivors and controls. Is there any differences between cancer treatments: Non-CNS vs CNS?

Response: We conducted additional formal analyses comparing the two survivor groups and found that indeed non-CNS survivors have higher EAA with the exception of PCGrimAge. These findings are reporting in Supplemental Table 2.

- Were the potential cognitive effects of non-CNS and CNS cancer therapies (such as chemotherapy and radiation) thoroughly accounted for in statistical models? For instance, analyses are adjusted for methotrexate dose but not for radiation nor other chemotherapies.

Response: All models were adjusted for cancer therapies recognized by the Children's Oncology Group Long-Term Follow Up Guidelines to increase the risk for long-term neurocognitive impairment. This includes cranial radiation, intrathecal chemotherapy, neurosurgery, high-dose methotrexate, and high-dose cytarabine. Any models among survivors treated with CNS-directed therapies were adjusted for all of these factors. Any models among survivors treated with non-CNS directed therapies were adjusted for high-dose methotrexate as the number of survivors treated with high-dose cytarabine was too small. We have reviewed the footnotes of each table to ensure accuracy.

- Related to my previous question, treatment can affect DNA methylation. There are some studies that have shown methylation sites affected by cancer treatments. Is there any overlap between sites affected by cancer treatment and these sites included in the epigenetic clocks? It would be very interesting since it would mean that treatment can alter methylation, and the methylation in these marks is maintained along this long-follow up period. I guess you do not have blood samples available at cancer diagnosis to actually check methylation maintenance.

Response: Unfortunately, we do not have biospecimens from diagnosis or immediately post treatment to understand the direct impact of cancer treatment in this sample or to understand the maintenance of any methylation changes. While it is well recognized that cancer treatments can impart methylation changes, we cannot speculate about the specific changes in our cohort and if any of those sites may be included in these clocks. In future work we hope to examine

CpG level specific associations to gain additional insights into the mechanisms linking specific cancer treatment exposure, DNA methylation level changes at single-CpG level and neurocognitive dysfunction.

- Regarding tables 3 and 4, are there any differences in the cognitive tests of the two survivors' groups separately or even combined compared to controls? It would be useful to see the comparison with the non-cancer individuals.

*Response: We have previously reported on the differences in cognitive test performance among those treated with and without CNS directed therapy (PMID: 33274777). Table 3 from this paper is included below for reference and includes the comparison between the two survivor groups, as well as a comparison to population norms (e.g. non-cancer, statistical significance indicated by *). As you can see for every neurocognitive outcome, the CNS-treated group performs significantly worse, even after adjustment for age at diagnosis, sex, and race and multiple comparisons indicating this group has reduced cognitive reserve that may make them more vulnerable to aging-related cognitive decline.*

On page 5 of the results we have added: "Analyses were stratified by receipt of CNS-directed therapy during treatment for childhood cancer as previous research has demonstrated this is associated with increased vulnerability to aging-related changes due to diminished cognitive reserve.[2, 38]"

We have clarified this in the methods on page 15: "Analyses were stratified by receipt of CNS-directed therapy during treatment for childhood cancer as previous research has demonstrated this is associated with increased vulnerability to aging-related changes due to diminished cognitive reserve.[2, 39]"

Williams et al (PMID: 33274777) Table 3: Comparison of CNS- and non-CNS-treated survivors on neurocognitive outcomes.

Neurocognitive Outcomes [^]	CNS-Treated (N=1597)		Non-CNS-Treated (N=1292)		P ²
	Mean (SD)	Impairment ¹ % (95% CI)	Mean (SD)	Impairment ¹ % (95% CI)	
Global Cognition					
Verbal reasoning	-0.471* (1.15)	21.64 (19.60, 23.68)	-0.223* (1.06)	15.04 (13.09,17.00)	<0.001
Non-Verbal reasoning	-0.116 (1.04)	12.03 (10.45, 13.75)	0.118* (0.91)	7.12 (5.77,8.67)	<0.001
Academics					
Word reading	-0.394* (0.77)	10.11 (8.63, 11.74)	-0.234* (0.63)	4.31 (3.17,5.44)	<0.001
Mathematics	-0.697* (1.10)	24.02 (21.88, 26.26)	-0.445* (0.92)	14.95 (12.95,16.95)	<0.001
Attention					
Sustained Attention	-0.327* (1.40)	16.70 (14.86, 18.67)	-0.052* (1.21)	10.33 (8.70,12.14)	<0.001
Variability	-0.352* (1.25)	17.74 (15.85,19.75)	-0.268 (1.20)	15.51 (13.55,17.63)	0.023
Commissions	-0.092 (1.11)	12.97 (11.32,14.76)	0.095* (1.04)	10.33 (8.70,12.14)	<0.001

	Focused attention	-0.252* (1.44)	17.99 (16.12,20.00)	0.240* (1.13)	8.95 (7.45,10.64)	<0.001
Processing speed						
	Visual-motor processing speed	-0.417* (1.14)	26.08 (23.91,28.34)	-0.018 (1.00)	12.13 (10.38,14.05)	<0.001
	Motor processing speed	-1.044* (1.52)	36.40 (34.01,38.85)	-0.584* (1.31)	23.24 (20.95,25.65)	<0.001
Memory						
	Short-term memory	-0.290* (1.03)	12.36 (10.77,14.10)	-0.042* (0.994)	7.38 (6.01,8.94)	<0.001
	Verbal Learning	-0.192* (1.27)	17.31 (15.46,19.28)	-0.005 (1.14)	12.68 (10.90,14.63)	<0.001
	Short-term verbal recall	-0.236* (1.27)	18.60 (16.69,20.62)	-0.082 (1.10)	13.23 (11.42,15.22)	<0.001
	Long-term verbal recall	-0.341* (1.30)	22.34 (20.29,24.50)	-0.201* (1.16)	18.20 (16.11,20.43)	<0.001
	Visual memory ³	-0.588* (1.22)	31.16 (28.66,33.75)	-0.251* (1.12)	21.60 (19.27,24.08)	<0.001
Executive function						
	Perseveration	-0.354* (1.36)	18.53 (16.61,20.57)	-0.163* (1.20)	14.61 (12.71, 16.69)	<0.001
	Working memory	-0.318* (0.97)	9.80 (8.37,11.38)	-0.156* (0.90)	4.20 (3.17,5.44)	<0.001
	Cognitive flexibility	-0.793* (1.71)	30.87 (28.58,33.23)	-0.262* (1.48)	20.34 (18.17,22.65)	<0.001
	Verbal fluency	-0.418* (1.16)	27.23 (25.03,29.51)	-0.166* (1.10)	17.77 (15.72,19.96)	<0.001
CNS: central nervous system. ^Higher scores for these measures are indicative of better functioning, * indicates mean is statistically significantly (P<0.05) different from population mean of 0 (SD=1). ¹ Impairment defined as having a Z-score below the 10 th percentile compared to national norms, ² linear regression models to compare neurocognitive performance in survivors treated with and without direct CNS-targeted therapies adjusted for age at diagnosis, sex, and race, corrected for the false discovery rate, ³ 15% of participants missing this test due to late addition to study protocol.						

- Given that telomere length (mLTL) was not significantly associated with neurocognition in the study, could the authors discuss why this might be the case, and whether telomere length remains a relevant biomarker for other aspects of aging in cancer survivors?

Response: We have added a paragraph on pages 11 and 12 of the discussion to highlight this: "Despite survivors of childhood cancer having significantly shorter telomeres[12], our data do not support an association between mLTL and neurocognitive dysfunction in long-term survivors of childhood cancer. This is consistent with one study of breast cancer survivors, 3-6 years post therapy[53], but in contrast to a study of breast cancer survivors immediately after treatment which reported shorter telomeres were associated with worse neurocognitive function in memory, attention, and executive function.[54] Telomere length may be a marker of acute damage and acute neurotoxicity, but does not represent the protracted aging-related pathways we aim to measure here. In line with this hypothesis, our previous work among childhood cancer survivors demonstrated that while survivors had shorter telomeres than non-cancer controls, this difference did not grow with age and showed a pattern of an accentuated aging. In contrast, epigenetic aging showed a pattern of accelerating aging with a steeper slope of annual change of epigenetic age comparing survivors with noncancer controls.[11, 12] This suggests that

telomeres may represent premature aging, or a truncation of physiological reserve after treatment, but do not represent accelerated aging, or continuing decline and therefore may not be associated with long-term neurocognitive impairment. Consistent with this hypothesis, we saw no overall effect of mLTL on verbal fluency or visual-motor processing speed, but when ALL survivors were stratified by age at diagnosis, among those diagnosed above the age of 10, higher mLTL residual was associated with better performance, but there was no association among those diagnosed under 10.” .

- Do the authors believe the biomarkers are specific to neurocognitive decline due to cancer treatment, or could these findings apply to other conditions with neurocognitive impact? Can you discuss more on this topic?

Response: The inspiration for this work came from aging research in the non-cancer general population, therefore, we agree with the reviewer that these findings have implications for other conditions with neurocognitive impact. It may, however, depend on the specific biomarker, some may be specific to cancer therapy that imparts a system wide pattern of molecular damage that is presumably sustained over time. Others may be specific to certain mechanisms and isolated damage as in traumatic brain injury. We do, however, agree that conditions (e.g. cardiovascular disease) that may increase the pace of biological aging through mechanisms like systemic inflammation, oxidative stress, and cellular senescence may be informed by these findings. In fact, in our data, depending on which clock was used to measure EAA, we saw different effects. Therefore, it may be that a new “clock” is needed that is trained to predict neurocognitive dysfunction.

We have added the following to the discussion on page 10 : “Our current data also suggest that PCGrimAge and DunedinPACE may be more sensitive than PCPhenoAge to neurocognitive dysfunction. In non-cancer populations, DunedinPACE and the original GrimAge clocks are highly correlated[33], and have been demonstrated to be better predictors of mortality than DNAmPhenoAge.[33, 49] In survivors of childhood cancer, GrimAge appears to be a slightly better predictor of frailty than DNAmPhenoAge[40], and we have demonstrated strong prospective associations between frailty and neurocognitive decline in this group.[51] Therefore, it remains unclear if these markers are surrogates for different pathological processes along the same causal pathway and additional work is needed to understand the functional implications of CpG sites included on each clock. Further, it may be that a new “clock” is needed that is trained to predict this early onset neurocognitive dysfunction among long-term survivors.”

- The study suggests the potential for EAA as a biomarker for intervention. Are there suggestions for how clinicians might use these biomarkers in practice? Could a risk stratification model based on EAA be feasibly implemented? Can you perform further analyses to see whether an optimal cut-point in EAA could be used to stratify those with impaired neurocognitive function and those with not?

Response: Currently, we do not feel we can make recommendations on how these biomarkers might be used in the clinical setting, as the technology is expensive and labor intensive and our findings have not yet been replicated. We do, however, feel that these may be useful in identifying those at risk of neurocognitive impairment who may benefit from intervention and subsequently measure the response to an intervention. In future longitudinal work we plan to examine the potential threshold for cut points to predict neurocognitive decline.

We have expanded the first paragraph of the discussion (page 9) to highlight potential research use: “Overall, these data suggest that epigenetic aging and neurocognitive aging may be closely linked and that EAA, specifically defined from PCGrimAge or DunedinPACE, may help identify those at greatest risk for neurocognitive impairment and who would benefit most from intervention. Additionally, given the modifiable nature of these biomarkers, they may hold utility in detecting pre-clinical changes in biological aging in response to interventions designed to improve physiologic and cognitive aging trajectories such as diet, physical activity, and senolytics.[43-45]”

Minor comments:

- Variables should be defined in full upon first mention, after which abbreviations may be used consistently.
- Add sample size in all tables.
- If you add data within the manuscript which is not shown in tables, mention “data not shown”
- Check supplementary figure 1 which last number might be wrong

Response: Thank you for pointing out these errors and omissions, we have defined all abbreviations, added sample sizes, and Supplemental Figure 1 has been updated.

- Can you briefly explain what the TelSeq method is?

*Response: TelSeq is a published computational method (PMID: 24609383) that we used for estimating the telomere length based on whole-genome sequencing data. It defines a sequence read as telomeric if it contains at least seven occurrences of the motif (TTAGGG). An estimate of telomere length is then computed by $tk * c/s$ where tk is the number of telomeric reads, c is a constant for genome length divided by 46 (i.e., number of telomere ends), and s is the total number of reads.*

- The paper would benefit from including at least one main figure to present the results in a more visual and accessible way

Response: We thank the reviewer for this suggestion, the previous table 2 has been converted into a figure to highlight differences in markers across groups.

We would like to thank the editor and reviewers for their additional consideration of our manuscript. We have included a point-point response below (in italics) as well as used track changes throughout the manuscript to address any remaining concerns or questions.

REVIEWER COMMENTS

Reviewer #1 (Remarks to the Author):

Authors have done an excellent job responding to all reviewer points.

Reviewer #2 (Remarks to the Author):

the authors have reasonably addressed the concerns I raised in my review. I have no further comments.

Reviewer #4 (Remarks to the Author):

Thank you for assessing most of my comments. The manuscript has improved but there are still two analyses that I would like to see:

- Causal mediation analyses to check whether methylation mediates the association between cancer treatment vs. controls and neurocognitive function.

Response: We have added the causal mediation analyses as requested, however, given the cross-sectional nature of these data, and the potential for residual confounding, we feel these analyses are exploratory in nature and the findings should be interpreted with certain caution.

We have added the following to the Results section on page 9: "Lastly, we conducted causal mediation analyses to examine if associations between treatments for childhood cancer and cognition were mediated by EAA, after adjusting for important confounders such as cancer treatments other than the one selected as the primary exposure, age at diagnosis, sex, BMI, physical activity, and smoking. These models were analyzed for any treatment, EAA, cognitive outcome path that met the following criteria: 1) the treatment has been previously demonstrated to be associated with worse neurocognitive function among survivors of childhood cancer[3], 2) in previously published work[11, 43] and our data, the treatment was associated with worse EAA, and 3) in our analyses, EAA was associated with worse neurocognitive functioning. Given the cross-sectional nature of these data, the collinearity between treatments, and the potential for residual confounding, we feel these results should be interpreted cautiously. Therefore, we are only reporting causal mediation results for paths where the total effect, the natural indirect effect, and the percentage mediated were statistically significant ($p < 0.05$, Supplemental Table 15). There were no statistically significant causal mediation paths among survivors treated without CNS-directed therapies. Among CNS-Treated survivors, DunedinPACE mediated associations between high-dose methotrexate and tests of global cognition (percent mediated [95%CI] = 21.7% [1.7, 41.7]) and attention (19.9% [0.3, 39.4]) accounting for

approximately 20% of the total effect. PCGrimAge mediated 28.8% of the association between chest radiation and word reading (28.8% [4.2, 53.4]).”

Supplemental Table 15: Statistically significant results from causal mediation analyses.

Treatment	EAA	Cognitive Outcome	Total Effect	p-value	Natural Direct Effect	p-value	Natural Indirect Effect	p-value	Percentage Mediated	p-value
CNS-Treated Survivors										
HD MTX	DunedinPACE	Verbal Reasoning	-0.28(-0.462,-0.099)	0.002	-0.219(-0.401,-0.038)	0.018	-0.061(-0.106,-0.015)	0.009	21.7(1.7,41.7)	0.034
HD MTX	DunedinPACE	Attention Variability	-0.285(-0.478,-0.092)	0.004	-0.228(-0.422,-0.035)	0.020	-0.057(-0.102,-0.012)	0.013	19.9(0.3,39.4)	0.046
Chest RT	PCGrimAge	Word Reading	0.251(0.055,0.448)	0.012	0.179(-0.009,0.367)	0.062	0.072(0.025,0.12)	0.003	28.8(4.2,53.4)	0.022

We have added the following to the discussion section on page 12: “... we have previously demonstrated that 935 CpG sites with substantial difference in DNA methylation across 538 genes are associated with a history of various childhood cancer treatments (e.g. alkylating agents, radiation) in this same sample of long-term survivors of childhood cancer.[57] Among these CpG sites, only 6 overlap with the CpG sites included on the clocks used in this analysis (Supplementary Table 16). The DNA methylation levels for other clock CpGs may have subtle treatment-related changes (i.e., cannot be detected individually at the epigenome-wide significance level) and contribute cumulatively to the overall observed difference in EAA between exposed and unexposed groups for specific cancer treatment. Further, the cause of cancer-related neurocognitive decline is likely multifactorial, and EAA may only account for a limited proportion of the causal pathway and other mechanisms such as DNA damage and inflammation should be explored.[58, 59] This is highlighted by the lack of mediation by EAA among the survivors treated without CNS-directed therapies. Among CNS-treated survivors, treatment associations with neurocognitive outcomes were only partially mediated by DunedinPACE and PCGrimAge. While this direct overlap between treatment-associated CpGs and epigenetic clock CpGs is limited, the composite DNA methylation alterations in all clock CpGs shows substantial difference reinforcing that cancer treatments influence EAA. Further work is needed to assess whether methylation changes at these loci are durable, directional, and whether they contribute meaningfully to accelerated epigenetic aging over long-term follow-up.”

We have added the following to the methods section on pages 17 and 18:” Lastly, we conducted causal mediation analyses[76, 77] to examine if associations between treatments for childhood cancer and cognition were mediated by EAA. These models were

analyzed for any treatment, EAA, cognitive outcome path that met the following criteria: 1) the treatment has been previously demonstrated to be associated with worse neurocognitive function among survivors of childhood cancer[3], 2) in previously published work[11, 43] and our data, the treatment was associated with worse EAA, and 3) in our analyses, EAA was associated with worse neurocognitive functioning. All causal mediation analyses were adjusted for potential confounders including age at diagnosis, sex, BMI, physical activity, and smoking. Chronic health conditions and education could not be adjusted for because they are directly impacted by treatment. Models among CNS-treated survivors were additionally adjusted for cranial radiation, intrathecal chemotherapy, high-dose methotrexate, and neurosurgery as appropriate (e.g. when that treatment is not the primary exposure). Given the cross-sectional nature of these data, the collinearity between treatments, and the potential for residual confounding, we feel these results should be interpreted cautiously. Therefore, we are only reporting causal mediation results for paths where the total effect, the natural indirect effect, and the percentage mediated were statistically significant ($p < 0.05$). “

- Any of the methylation sites included in clocks is affected by cancer treatment in previous studies? Perform an overlap with findings from previous studies in the literature

Response: Our previous work has demonstrated that 935 CpG sites were associated with a history of several different cancer treatments in these very long-term survivors from the St. Jude Lifetime Cohort (PMID: 33823916). Of these, only 6 were found to be part of the clocks used in this analysis (see table below), suggesting that there was neither enrichment nor depletion of treatment-associated CpGs among the clock CpGs. Nonetheless, these findings provided direct evidence showing that some treatment-associated CpG sites are indeed part of aging clocks. Future work is needed to determine whether treatment-associated methylation variations at these CpG sites are indeed caused by treatment exposures using paired blood samples of pre-treatment and immediately post-treatment and are durable based on serial samples collected over the long follow-up. Once further established, these methylation sites may provide potential targets for interventions.

We have added the following to the discussion section on page 12: “Several studies have demonstrated that chemotherapy and radiation can impart DNA methylation changes at many different CpG sites among patients with breast and gastric cancer.[25, 27, 54-56] If these changes are sustained over time, it may offer important insights into how cancer treatments might influence biological aging trajectories and increase the risk for neurocognitive impairment. While we do not have access to pre-treatment and immediately post-treatment blood samples from our cohort to directly assess the persistence of methylation changes, we have previously demonstrated that 935 CpG sites with substantial difference in DNA methylation across 538 genes are associated with a history of various childhood cancer treatments (e.g. alkylating agents, radiation) in this same sample of long-term survivors of childhood cancer.[57] Among these CpG sites, only 6 overlap with the CpG sites included on the clocks used in this analysis (Supplementary Table 16). The DNA methylation levels for other clock CpGs may have subtle treatment-related changes (i.e., cannot be detected individually at the epigenome-

wide significance level) and contribute cumulatively to the overall observed difference in EAA between exposed and unexposed groups for specific cancer treatment. Further, the cause of cancer-related neurocognitive decline is likely multifactorial, and EAA may only account for a limited proportion of the causal pathway and other mechanisms such as DNA damage and inflammation should be explored.[58, 59] This is highlighted by the lack of mediation by EAA among the survivors treated without CNS-directed therapies. Among CNS-treated survivors, treatment associations with neurocognitive outcomes were only partially mediated by DunedinPACE and PCGrimAge. While this direct overlap between treatment-associated CpGs and epigenetic clock CpGs is limited, the composite DNA methylation alterations in all clock CpGs shows substantial difference reinforcing that cancer treatments influence EAA. Further work is needed to assess whether methylation changes at these loci are durable, directional, and whether they contribute meaningfully to accelerated epigenetic aging over long-term follow-up.”

Supplemental Table 16: Overlap of differentially methylated CpG sites associated with treatment for childhood cancer and CpG sites included in epigenetic aging clocks.

Treatment	CpG	HGNCgene	PCHorvath	PCHannum	PCPheno	DunedinPACE	GrimAge ¹
Chest-RT	cg05316065	GSDMC	FALSE	FALSE	TRUE	FALSE	NA
Epipodophyllotoxins	cg06738602	PTGER2	TRUE	FALSE	FALSE	FALSE	NA
Alkylating agents	cg14200569	PRDM16	FALSE	FALSE	FALSE	TRUE	NA
Antimetabolites	cg14200569	PRDM16	FALSE	FALSE	FALSE	TRUE	NA
Asparaginase enzymes	cg14200569	PRDM16	FALSE	FALSE	FALSE	TRUE	NA
Epipodophyllotoxins	cg14200569	PRDM16	FALSE	FALSE	FALSE	TRUE	NA
Abdomen-RT	cg14200569	PRDM16	FALSE	FALSE	FALSE	TRUE	NA
Chest-RT	cg14200569	PRDM16	FALSE	FALSE	FALSE	TRUE	NA
Pelvic-RT	cg14200569	PRDM16	FALSE	FALSE	FALSE	TRUE	NA
Abdomen-RT	cg17061862	NA	FALSE	FALSE	FALSE	TRUE	NA
Abdomen-RT	cg19283806	CCDC102B	FALSE	TRUE	FALSE	FALSE	NA
Chest-RT	cg19283806	CCDC102B	FALSE	TRUE	FALSE	FALSE	NA
Pelvic-RT	cg19283806	CCDC102B	FALSE	TRUE	FALSE	FALSE	NA
Abdomen-RT	cg26581729	NPDC1	FALSE	FALSE	TRUE	FALSE	NA

¹ Catalog of CpG sites included on PC GrimAge is not currently publicly available.